# A Reliable Cryptographic Framework for Empirical Machine Unlearning Evaluation

**Yiwen Tu**[*]
University of Michigan, Ann Arbor
evantu@umich.edu

**Pingbang Hu**[*]
University of Illinois Urbana-Champaign
pbb@illinois.edu

**Jiaqi W. Ma**
University of Illinois Urbana-Champaign
jiaqima@illinois.edu

## Abstract

Machine unlearning updates machine learning models to remove information from specific training samples, complying with data protection regulations that allow individuals to request the removal of their personal data. Despite the recent development of numerous unlearning algorithms, reliable evaluation of these algorithms remains an open research question. In this work, we focus on membership inference attack (MIA) based evaluation, one of the most common approaches for evaluating unlearning algorithms, and address various pitfalls of existing evaluation metrics lacking theoretical understanding and reliability. Specifically, by modeling the proposed evaluation process as a *cryptographic game* between unlearning algorithms and MIA adversaries, the naturally induced evaluation metric measures the data removal efficacy of unlearning algorithms and enjoys provable guarantees that existing evaluation metrics fail to satisfy. Furthermore, we propose a practical and efficient approximation of the induced evaluation metric and demonstrate its effectiveness through both theoretical analysis and empirical experiments. Overall, this work presents a novel and reliable approach to empirically evaluating unlearning algorithms, paving the way for the development of more effective unlearning techniques.

## 1 Introduction

*Machine unlearning* is an emerging research field in artificial intelligence (AI) motivated by the "Right to be Forgotten," outlined by various data protection regulations such as the General Data Protection Regulation (GDPR) [Mantelero, 2013] and the California Consumer Privacy Act (CCPA) [CCPA, 2018]. Specifically, the Right to be Forgotten grants individuals the right to request that an organization erase their personal data from its databases, subject to certain exceptions. Consequently, when such data were used for training machine learning models, the organization may be required to update their models to "unlearn" the data to comply with the Right to be Forgotten. A naive solution is retraining the model on the remaining data after removing the requested data points, but this solution is computationally prohibitive. Recently, a plethora of unlearning algorithms have been developed to efficiently update the model without complete retraining, albeit usually at the price of removing the requested data information only approximately [Cao and Yang, 2015, Bourtoule et al., 2021, Guo et al., 2020, Neel et al., 2021, Sekhari et al., 2021, Chien et al., 2023, Kurmanji et al., 2023].

---

[*]Equal contribution.

39th Conference on Neural Information Processing Systems (NeurIPS 2025).

Despite the active development of unlearning algorithms, the fundamental problem of properly evaluating these methods remains an open research question, as highlighted by the Machine Unlearning Competition held at NeurIPS 2023[2]. The unlearning literature has developed a variety of evaluation metrics for measuring the *data removal efficacy* of unlearning algorithms, i.e., to which extent the information of the requested data points are removed from the unlearned model. Existing metrics can be roughly categorized as attack-based [Graves et al., 2020, Kurmanji et al., 2023, Goel et al., 2023, Hayes et al., 2024, Sommer et al., 2020, Goel et al., 2023], theory-based [Triantafillou and Kairouz, 2023, Becker and Liebig, 2022], and retraining-based [Golatkar et al., 2021, Wu et al., 2020, Izzo et al., 2021], respectively. Each metric has its own limitations and there is no consensus on a standard evaluation metric for unlearning. Among these metrics, the membership inference attack (MIA) based metric, which aims to determine whether specific data points were part of the original training dataset based on the unlearned model, is perhaps the most commonly seen in the literature. MIA is often considered a natural unlearning evaluation metric as it directly measures the privacy leakage of the unlearned model, which is a primary concern of unlearning algorithms.

Most existing literature directly uses MIA performance[3] to measure the data removal efficacy of unlearning algorithms. However, such metrics can be unreliable as MIA performance is not a well-calibrated metric when used for unlearning evaluation, leading to counterintuitive results. For example, naively retraining the model is theoretically optimal for data removal efficacy, albeit computationally prohibitive. Nevertheless, retraining is not guaranteed to yield the lowest MIA performance compared to other approximate unlearning algorithms. This discrepancy arises because MIAs themselves are imperfect and can make mistakes in inferring data membership. Furthermore, MIA performance is also sensitive to the composition of data used to conduct MIA and the specific choice of MIA algorithm. Consequently, the results obtained using different MIAs are not directly comparable and can vary significantly, making it difficult to draw definitive conclusions about the efficacy of unlearning algorithms. These limitations render the existing MIA-based evaluation brittle and highlight the need for a more reliable and comprehensive framework for assessing the performance of unlearning algorithms.

In this work, we aim to address the challenges associated with MIA-based unlearning evaluation by introducing a game-theoretical framework named the *unlearning sample inference game*. Within this framework, we gauge the data removal efficacy through a game where, informally, the challenger (model provider) endeavors to produce an unlearned model, while the adversary (MIA adversary) seeks to exploit the unlearned model to determine the membership status of the given samples. By carefully formalizing the game, with controlled knowledge and interaction between both parties, we ensure that the success rate of the adversary in the unlearning sample inference game possesses several desirable properties, and thus can be used as an unlearning evaluation metric, circumventing the aforementioned pitfalls of MIA performance. Specifically, it ensures that the adversary's success rate towards the retrained model is precisely zero, thereby certifying retraining as the theoretically optimal unlearning method. Moreover, it provides a provable guarantee for certified machine unlearning algorithms [Guo et al., 2020], aligning the proposed metric with theoretical results in the literature. Lastly, it inherently accommodates the existence of multiple MIA adversaries, resolving the conflict between different choices of MIAs. However, the computational demands of exactly calculating the proposed metric pose a practical issue. To mitigate this, we introduce a *SWAP* test as a practical approximation, which also inherits many of the desirable properties of the exact metric. Empirically, this test proves robust to changes in random seed and dataset size, enabling model maintainers to conduct small-scale experiments to gauge the quality of their unlearning algorithms.

Finally, we highlight our contributions in this work as follows:

- We present a formalization of the *unlearning sample inference game*, establishing a novel unlearning evaluation metric for data removal efficacy.
- We demonstrate several provable properties of the proposed metric, circumventing various pitfalls of existing MIA-based metrics.
- We introduce a straightforward and effective *SWAP* test for efficient empirical analysis. Through thorough theoretical examination and empirical experiments, we show that it exhibits similar desirable properties.

---

[2]See `https://unlearning-challenge.github.io/`.

[3]For example, the accuracy or the area under the receiver operating characteristic curve (AUC) of the inferred membership.

In summary, this work offers a game-theoretic framework for reliable empirical evaluation of machine unlearning algorithms, tackling one of the most foundational problems in this field.

## 2 Related work

### 2.1 Machine unlearning

Machine unlearning, as initially introduced by Cao and Yang [2015], is to update machine learning models to remove the influence of selected training data samples, effectively making the models "forget" those samples. Most unlearning methods can be categorized as exact unlearning and approximate unlearning. Exact unlearning requires the unlearned models to be indistinguishable from models that were trained from scratch without the removed data samples. However, it can still be computationally expensive, especially for large datasets and complex models. On the other hand, approximate unlearning aims to remove the influence of selected data samples while accepting a certain level of deviation from the exactly unlearned model. This allows for more efficient unlearning algorithms, making approximate unlearning increasingly popular practically. While approximate unlearning is more time and space-efficient, it does not guarantee the complete removal of the influence of the removed data samples. We refer the audience to the survey on unlearning methods by Xu et al. [2023] for a more comprehensive overview.

### 2.2 Machine unlearning evaluation

Evaluating machine unlearning involves considerations of computational efficiency, model utility, and data removal efficacy. Computational efficiency refers to the time and space complexity of the unlearning algorithms, while model utility measures the prediction performance of the unlearned models. These two aspects can be measured relatively straightforwardly through, e.g., computation time, memory usage, or prediction accuracy. Data removal efficacy, on the other hand, assesses the extent to which the influence of the requested data points has been removed from the unlearned models, which is highly non-trivial to measure and has attracted significant research efforts recently. These efforts for evaluating or guaranteeing data removal efficacy can be categorized into several groups. We provide an overview below and refer readers to Appendix A for an in-depth review.

- **Retraining-based**: Generally, retraining-based evaluation measures the parameter or posterior difference between unlearned models and retrained models, the gold standard for data removal [Golatkar et al., 2020, 2021, He et al., 2021, Izzo et al., 2021, Peste et al., 2021, Wu et al., 2020]. However, they are often unreliable as measures like parameter difference can be sensitive to the randomness led by the training dynamics [Cretu et al., 2023].

- **Theory-based**: Another line of work tries to characterize data removal efficacy by requiring a strict theoretical guarantee for the unlearned models [Chien et al., 2023, Guo et al., 2020, Neel et al., 2021] or turning to information-theoretic analysis [Becker and Liebig, 2022]. However, they have strong model assumptions or require inefficient white-box access to target models, thus limiting their applicability in practice.

- **Attack-based**: Since attacks are the most direct way to interpret privacy risks, attack-based evaluation is a common metric in unlearning literature [Chen et al., 2021, Goel et al., 2023, Graves et al., 2020, Hayes et al., 2024, Kurmanji et al., 2023, Sommer et al., 2020, Song and Mittal, 2021]. Our work belongs to this category and addresses the pitfalls of existing attack-based methods.

## 3 Proposed evaluation framework

### 3.1 Preliminaries

To mitigate the limitations of directly using MIA accuracy as the evaluation metric for unlearning algorithms, we draw inspiration from *cryptographic games* [Katz and Lindell, 2007], which are a fundamental tool to define and analyze the security properties of cryptographic protocols. In particular, we leverage the notion of ***advantage*** to form a more reliable and well-calibrated metric for evaluating the effectiveness of unlearning algorithms. We leave a more detailed description to Appendix B.1.

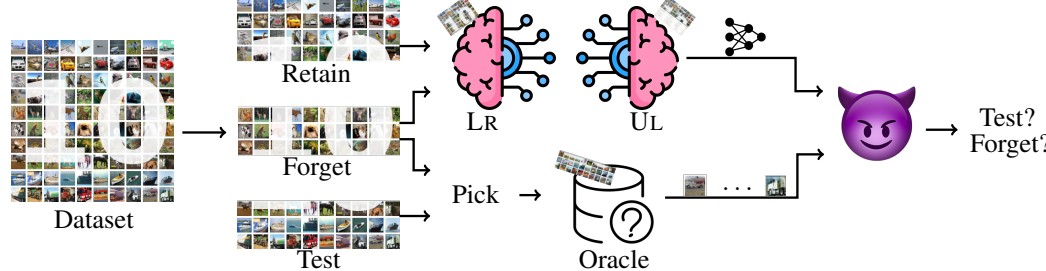

Figure 1: The unlearning sample inference game framework for our machine unlearning evaluation.

In the rest of this section, we introduce the proposed unlearning evaluation framework based on a carefully designed cryptographic game and an advantage metric associated with the game. We also provide provable guarantees for the soundness of the proposed metric.

## 3.2 Unlearning sample inference game

We propose the *unlearning sample inference game* $\mathcal{G} = (\text{UL}, \mathcal{A}, \mathcal{D}, \mathbb{P}_{\mathcal{D}}, \alpha)$ that characterizes the privacy risk of the unlearned models against an MIA adversary. It involves two players (a *challenger* named UL and an *adversary* $\mathcal{A}$), a finite dataset $\mathcal{D}$ with a *sensitivity distribution* $\mathbb{P}_{\mathcal{D}}$ defined over $\mathcal{D}$, and an *unlearning portion* parameter $\alpha$. Intuitively, the game works as follows:

- the challenger UL performs unlearning on a "forget set" of data for a model trained on the union of the "retain set" and the "forget set," with sizes of two subsets of $\mathcal{D}$ subject to a ratio $\alpha$;

- the adversary $\mathcal{A}$ attacks the challenger's unlearned model by telling whether some random data points (according to $\mathbb{P}_{\mathcal{D}}$) are originally in the "forget set" or an unused set of data called "test set."

An illustration is given in Figure 1. A detailed discussion of various design choices we made can be found in Appendix B.2. Below, we formally introduce the game, starting with the initialization phase.

**Initialization.** The game starts by randomly splitting the dataset $\mathcal{D}$ into three disjoint sets: a *retain set* $\mathcal{R}$, a *forget set* $\mathcal{F}$, and a *test set* $\mathcal{T}$, i.e., $\mathcal{D} =: \mathcal{R} \cup \mathcal{F} \cup \mathcal{T}$, subject to the following restrictions:

(a) $\alpha = |\mathcal{F}|/|\mathcal{R} \cup \mathcal{F}|$: The *unlearning portion* $\alpha$ specifies how much data needs to be unlearned with respect to the original dataset used by the model.

(b) $|\mathcal{F}| = |\mathcal{T}|$: The sizes of $\mathcal{F}$ and $\mathcal{T}$ are equal to avoid potential inductive biases.

Under restrictions (a) and (b), the size of $\mathcal{R}$, $\mathcal{F}$, and $\mathcal{T}$ are determined, depending on $\alpha$. We denote $\mathcal{S}_{\alpha}$ as the finite collection of all possible dataset splits satisfying restriction (a) and (b) such that $s \sim \mathcal{U}(\mathcal{S}_{\alpha})$[4] is in the form of $s = (\mathcal{R}, \mathcal{F}, \mathcal{T})$, where the tuple is ordered by the retain, forget, and test set. After splitting $\mathcal{D}$ according to $s$, a *random oracle* $\mathcal{O}_s(b)$ is then constructed according to $s$ and the sensitivity distribution $\mathbb{P}_{\mathcal{D}}$, together with a secret bit $b \in \{0, 1\}$. The intuition of this random oracle is that it offers the "two scenarios" we mentioned in Section 3.1, respectively specified by $b = 0$ and $b = 1$: when the oracle $\mathcal{O}_s(b)$ is called, it emits a data point $x \sim \mathcal{O}_s(b)$ sampled from either $\mathcal{F}$ (when $b = 0$) or $\mathcal{T}$ (when $b = 1$), where the sampling probability is respect to $\mathbb{P}_{\mathcal{D}}$.

We make some remarks on the role of the sensitivity distribution $\mathbb{P}_{\mathcal{D}}$, as it seems opaque at first glance. Intuitively, $\mathbb{P}_{\mathcal{D}}$ captures biases stemming from various origins, such that more sensitive data will have greater sampling probability, hence greater privacy risks. For instance, if the forget set comprises data that users request to delete, with some being more sensitive than others, a corresponding bias should be incorporated into the game. In particular, we tailor our random oracle $\mathcal{O}_s(b)$ to sample data according to $\mathbb{P}_{\mathcal{D}}$, so when the adversary engages with the oracle, it gains increased exposure to more sensitive data, compelling the challenger to unlearn such data more effectively, thereby necessitating a heightened level of defense.

**Challenger Phase.** The challenger is given the retain set $\mathcal{R}$, the forget set $\mathcal{F}$, and a learning algorithm LR (takes a dataset and outputs a learned model) and a unlearning algorithm UL (takes the

---

[4]Throughout the paper, $\mathcal{U}(\cdot)$ denotes the uniform distribution.

original model and a training subset to unlearn, and outputs an unlearned model). For simplicity, we denote the challenger as UL, as the unlearning algorithm is the component under evaluation. The goal of the challenger is to unlearn $\mathcal{F}$ by UL from the model trained with LR on $\mathcal{R} \cup \mathcal{F}$. Intuitively, for an ideal UL, for any $x \in \mathcal{F} \cup \mathcal{T}$, it is statistically impossible to decide whether $x \in \mathcal{F}$ or $x \in \mathcal{T}$ given accesses to the unlearned model $m := \mathrm{UL}(\mathrm{LR}(\mathcal{R} \cup \mathcal{F}), \mathcal{F})$. As both LR and UL can be randomized, $m$ follows a distribution $\mathbb{P}_{\mathcal{M}}(\mathrm{UL}, s)$ depending on the split $s$ and UL, where $\mathcal{M}$ denotes the set of all possible models. This distribution $\mathbb{P}_{\mathcal{M}}(\mathrm{UL}, s)$ summarizes the result of the challenger.

**Adversary Phase.** The adversary $\mathcal{A}$ is an (efficient) algorithm that has access to the unlearned model $m = \mathrm{UL}(\mathrm{LR}(\mathcal{R} \cup \mathcal{F}), \mathcal{F})$ and the random oracle $\mathcal{O} = \mathcal{O}_s(b)$, where both $s$ and $b$ are unknown to $\mathcal{A}$. The goal of the adversary is to guess $b \in \{0, 1\}$ by interacting with $m$ and $\mathcal{O}$, i.e., after interacting with $\mathcal{O}$ and $m$, decide whether the data points from $\mathcal{O}$ are from $\mathcal{F}$ or $\mathcal{T}$. Notation-wise, we write $\mathcal{A}^{\mathcal{O}}(m) \mapsto \{0, 1\}$. Note that in one play of the game, $\mathcal{O}$ is fixed as either $\mathcal{O}_s(0)$ or $\mathcal{O}_s(1)$ but will not switch between $b = 0$ and $b = 1$.

## 3.3 Advantage and unlearning quality

By viewing the unlearning sample inference game as a cryptographic game, with the discussion in Section 3.1, the corresponding *advantage* can be defined as follows:

**Definition 3.1** (Advantage). *Given an unlearning sample inference game $\mathcal{G} = (\mathrm{UL}, \mathcal{A}, \mathcal{D}, \mathbb{P}_{\mathcal{D}}, \alpha)$, the* advantage *of $\mathcal{A}$ against* UL *is defined as*

$$\mathrm{Adv}(\mathcal{A}, \mathrm{UL}) = \frac{1}{|\mathcal{S}_\alpha|} \left| \sum_{\substack{s \in \mathcal{S}_\alpha \\ \mathcal{O} = \mathcal{O}_s(0)}} \mathrm{Pr}_{\substack{m \sim \mathbb{P}_{\mathcal{M}}(\mathrm{UL}, s)}}(\mathcal{A}^{\mathcal{O}}(m) = 1) \quad - \sum_{\substack{s \in \mathcal{S}_\alpha \\ \mathcal{O} = \mathcal{O}_s(1)}} \mathrm{Pr}_{\substack{m \sim \mathbb{P}_{\mathcal{M}}(\mathrm{UL}, s)}}(\mathcal{A}^{\mathcal{O}}(m) = 1) \right|.$$

To simplify notation, we sometimes omit $m \sim \mathbb{P}_{\mathcal{M}}(\mathrm{UL}, s)$ and substitute $\mathcal{O}_s(b)$ to the superscript of $\mathcal{A}$ when it's clear, i.e., we can write $\mathrm{Pr}(\mathcal{A}^{\mathcal{O}_s(b)}(m) = 1)$. With the definition of advantage, measuring the quality of the challenger UL is standard by considering the worst-case guarantee:

**Definition 3.2** (Unlearning Quality). *For any unlearning algorithm* UL*, its* Unlearning Quality *under an unlearning sample inference game $\mathcal{G}$ is defined as*

$$\mathcal{Q}(\mathrm{UL}) := 1 - \sup_{\mathcal{A}} \mathrm{Adv}(\mathcal{A}, \mathrm{UL}),$$

*where the supermum is over all efficient adversary $\mathcal{A}$.*

Our definition of advantage (Unlearning Quality) has several theoretical merits as detailed below.

**Zero Grounding for** RETRAIN**.** Consider UL being the gold-standard unlearning method, i.e., the retraining method RETRAIN where $\mathrm{RETRAIN}(\mathrm{LR}(\mathcal{R} \cup \mathcal{F}), \mathcal{F}) = \mathrm{LR}(\mathcal{R})$. Since the forget set $\mathcal{F}$ and the test set $\mathcal{T}$ are all unforeseen data to retrained models trained only on $\mathcal{R}$, one should expect RETRAIN to defend any adversary $\mathcal{A}$ perfectly, leading to a zero advantage. The following Theorem 3.3 shows that our definition of advantage in Definition 3.1 indeed achieves such a desirable zero grounding property.

**Theorem 3.3** (Zero Grounding). *For any adversary $\mathcal{A}$, $\mathrm{Adv}(\mathcal{A}, \mathrm{RETRAIN}) = 0$ where* RETRAIN *is the retraining method. Hence, $\mathcal{Q}(\mathrm{RETRAIN}) = 1$.*

The proof of Theorem 3.3 can be found in Appendix B.3.

At a high level, the zero grounding property of the advantage is due to its symmetry—we measure the difference between $\mathcal{O}_s(0)$ and $\mathcal{O}_s(1)$ across all possible splits in $\mathcal{S}_\alpha$, such that each data point has the same chance to appear in both the forget set $\mathcal{F}$ and the test set $\mathcal{T}$. In comparison to conventional MIA-based evaluation that only measures the MIA performance on a single data split, this symmetry guarantees that all MIA adversaries have a zero advantage on RETRAIN even if the MIA is biased for certain data points, as the bias will be canceled out between symmetric splits that put these data points in $\mathcal{F}$ and $\mathcal{T}$ respectively.

**Guarantee Under Certified Removal.** We establish an upper bound on the advantage using the well-established notion of *certified removal* [Guo et al., 2020], which is inspired by differential privacy [Dwork, 2006]:

**Definition 3.4** (Certified Removal [Guo et al., 2020]; Informal). *For a fixed dataset $\mathcal{D}$, let* LR *and* UL *be learning & unlearning algorithm respectively, and denote $\mathcal{H}$ to be the hypothesis class containing all possible models that can be produced by* LR *and* UL*. Then, for any $\epsilon, \delta > 0$, the unlearning algorithm* UL *is said to be $(\epsilon, \delta)$-certified removal if for any $\mathcal{W} \subset \mathcal{H}$ and for **any** disjoint $\mathcal{R}, \mathcal{F} \subseteq \mathcal{D}$*

$$\begin{cases} \Pr(\text{UL}(\text{LR}(\mathcal{R} \cup \mathcal{F}), \mathcal{F}) \in \mathcal{W}) \leq e^\epsilon \Pr(\text{RETRAIN}(\text{LR}(\mathcal{R} \cup \mathcal{F}), \mathcal{F}) \in \mathcal{W}) + \delta; \\ \Pr(\text{RETRAIN}(\text{LR}(\mathcal{R} \cup \mathcal{F}), \mathcal{F}) \in \mathcal{W}) \leq e^\epsilon \Pr(\text{UL}(\text{LR}(\mathcal{R} \cup \mathcal{F}), \mathcal{F}) \in \mathcal{W}) + \delta. \end{cases}$$

Given its root in differential privacy[5], certified removal has been widely accepted as a rigorous measure of the goodness of approximate unlearning methods [Neel et al., 2021, Chien et al., 2023], where smaller $\epsilon$ and $\delta$ indicate better unlearning. However, in practice, it is difficult to empirically quantify $(\epsilon, \delta)$ for most approximate unlearning methods.

In the following Theorem 3.5, we provide a lower bound for the proposed Unlearning Quality for an $(\epsilon, \delta)$-certified removal unlearning algorithm, showing the close theoretical connection between the proposed Unlearning Quality metric and certified removal, while the proposed metric is easier to measure empirically.

**Theorem 3.5** (Guarantee Under Certified Removal). *Given an $(\epsilon, \delta)$-certified removal unlearning algorithm* UL *with some $\epsilon, \delta > 0$, for any adversary $\mathcal{A}$ against* UL*, we have* $\text{Adv}(\mathcal{A}, \text{UL}) \leq 2 \cdot (1 - \frac{2 - 2\delta}{e^\epsilon + 1})$. *Hence,* $\mathcal{Q}(\text{UL}) \geq \frac{4 - 4\delta}{e^\epsilon + 1} - 1$.

The formal definition of certified removal and the proof of Theorem 3.5 can be found in Appendix B.4.

### 3.4 *SWAP* test

Direct calculation requires enumerating dataset splits in $\mathcal{S}_\alpha$, which is computationally infeasible. Hence, we propose a simple approximation scheme named the *SWAP* test, which requires as few as two dataset splits to approximate the advantage and still preserves desirable properties as the original definition. The idea is to consider the *swap pair* between a forget set $\mathcal{F}$ and a test set $\mathcal{T}$. Specifically, pick a random split $s = (\mathcal{R}, \mathcal{F}, \mathcal{T}) \in \mathcal{S}_\alpha$ and calculate the term corresponding to $s$ in Definition 3.1:

$$\text{Adv}_s(\mathcal{A}, \text{UL}) := \Pr_{m \sim \mathbb{P}_\mathcal{M}(\text{UL}, s)}(\mathcal{A}^{\mathcal{O}_s(0)}(m) = 1) - \Pr_{m \sim \mathbb{P}_\mathcal{M}(\text{UL}, s)}(\mathcal{A}^{\mathcal{O}_s(1)}(m) = 1).$$

Next, *swap* $\mathcal{F}$ and $\mathcal{T}$ in $s$ to get $s' = (\mathcal{R}, \mathcal{T}, \mathcal{F})$, and calculate its corresponding term in Definition 3.1:

$$\text{Adv}_{s'}(\mathcal{A}, \text{UL}) := \Pr_{m \sim \mathbb{P}_\mathcal{M}(\text{UL}, s')}(\mathcal{A}^{\mathcal{O}_{s'}(0)}(m) = 1) - \Pr_{m \sim \mathbb{P}_\mathcal{M}(\text{UL}, s')}(\mathcal{A}^{\mathcal{O}_{s'}(1)}(m) = 1).$$

Finally, average the two advantages above and obtain

$$\overline{\text{Adv}}_{\{s, s'\}}(\mathcal{A}, \text{UL}) := \frac{\left| \text{Adv}_s(\mathcal{A}, \text{UL}) + \text{Adv}_{s'}(\mathcal{A}, \text{UL}) \right|}{2}.$$

In essence, we approximate Definition 3.1 by replacing $\mathcal{S}_\alpha$ with $\{s, s'\}$. Note that the *SWAP* test relies on the restriction (b) to be valid, i.e., $|\mathcal{F}| = |\mathcal{T}|$.

***SWAP* Test versus Random Splits.** The key insight is that the *SWAP* test reserves the symmetry in the original definition of advantage, and as shown in Proposition 3.6 (see Remark B.2 for proof), it still grounds the advantage of any adversary $\mathcal{A}$ against RETRAIN to zero, preserving the same theoretical guarantees as Theorem 3.3.

**Proposition 3.6** (Zero Grounding of *SWAP* Test (Informal)). *For any adversary $\mathcal{A}$ and swap splits $s, s' \in \mathcal{S}_\alpha$, $\overline{\text{Adv}}_{\{s, s'\}}(\mathcal{A}, \text{RETRAIN}) = 0$.*

On the contrary, naively taking two random splits with non-empty overlap can lead to an adversary with high advantage against RETRAIN:

**Proposition 3.7** (High Advantage Under Random Splits). *For any two splits $s_1, s_2 \in \mathcal{S}_\alpha$ satisfying a moderate non-degeneracy assumption, there's an efficient deterministic adversary $\mathcal{A}$ such that $\overline{\text{Adv}}_{\{s_1, s_2\}}(\mathcal{A}, \text{UL}) = 1$ for **any** unlearn method* UL*. Particularly, $\overline{\text{Adv}}_{\{s_1, s_2\}}(\mathcal{A}, \text{RETRAIN}) = 1$.*

---

[5]In fact, it has been shown that a model with differential privacy guarantees automatically enjoys certified removal guarantees for any training data point [Guo et al., 2020].

The full statement and the proof of Proposition 3.7 can be found in Appendix B.5.

**Remark 3.8** (Offsetting MIA Accuracy/AUC for RETRAIN). *One may wonder whether we could achieve zero grounding by simply offsetting the MIA accuracy for RETRAIN to zero (or offsetting the MIA AUC to 0.5). In Appendix B.6, we provide a discussion on why this strategy will lead to pathological cases for measuring unlearning performance.*

### 3.5 Practical implementation

While the proposed *SWAP* test significantly reduces the computational cost for evaluating the advantage of an adversary, evaluating the Unlearning Quality is still challenging since: 1.) most of the state-of-the-art MIAs do not exploit the covariance between data points; 2.) it is impossible to solve the supremum in Definition 3.2 exactly. We will start by addressing the first challenge.

**Weak Adversary.** As the current state-of-the-art MIAs make independent decisions on each data point [Bertran et al., 2023, Shokri et al., 2017, Carlini et al., 2022] without considering their covariance, therefore, for empirical analysis, we accommodate our unlearning sample inference game by restricting the adversary's knowledge such that it can only interact with the oracle once. We call such a adversary as *weak adversary* $\mathcal{A}_\mathrm{w}$, which will first learn a binary classifier $f(\cdot)$ by interacting with $m$, and output its prediction of $b$ as $f(x)$ by querying the oracle $\mathcal{O} = \mathcal{O}_s(b)$ *exactly once* to obtain $x \sim \mathcal{O}$, where both $s$ and $b$ are unknown to $\mathcal{A}_\mathrm{w}$. In this case, its advantage can be defined as

$$\mathrm{Adv}(\mathcal{A}_\mathrm{w}, \mathrm{UL}) = \frac{1}{|\mathcal{S}_\alpha|} \left| \sum_{s \in \mathcal{S}_\alpha} \Pr_{\substack{m \sim \mathbb{P}_\mathcal{M}(\mathrm{UL},s) \\ x \sim \mathcal{O}_s(0)}} (\mathcal{A}_\mathrm{w}(m,x)=1) - \sum_{s \in \mathcal{S}_\alpha} \Pr_{\substack{m \sim \mathbb{P}_\mathcal{M}(\mathrm{UL},s) \\ x \sim \mathcal{O}_s(1)}} (\mathcal{A}_\mathrm{w}(m,x)=1) \right|$$

and the Unlearning Quality now becomes $\mathcal{Q} := 1 - \sup_{\mathcal{A}_\mathrm{w}} \mathrm{Adv}(\mathcal{A}_\mathrm{w}, \mathrm{UL})$, analogously to Definitions 3.1 and 3.2. These new definitions are subsumed under the original paradigm since the only difference is the number of interactions with the oracle.

**Approximating the Supremum.** While it is impossible to solve the supremum in Definition 3.2 exactly, a plausible interpretation is that the supremum is *approximately* solved by the adversary, as most of the state-of-the-art MIA adversaries are formulated as end-to-end optimization problems [Bertran et al., 2023]. By assuming these MIA adversaries are trying to maximize the advantage when constructing the final classifier $f(\cdot)$ and that the search space is large enough to parameterize all the possible weak adversaries of our interests, we can interpret that the supermum is approximately solved. Moreover, in practice, one can refine the estimation of the supremum by selecting the most potent among multiple state-of-the-art MIA adversaries.

## 4 Experiment

In this section, we provide empirical evidence of the effectiveness of the proposed evaluation framework. In what follows, for brevity, we will use *SWAP* test to refer to the proposed practical approximations for calculating the proposed evaluation metric, which in reality is a combination of the *SWAP* test in Section 3.4 and other approximations discussed in Section 3.5. We further denote $\mathcal{Q}$ as the proposed metric, Unlearning Quality, calculated by the *SWAP* test. With these notations established, our goal is to validate the theoretical results, demonstrate additional observed benefits of the proposed Unlearning Quality metric, and ultimately show that it outperforms other attack-based evaluation metrics. More details can be found in Appendix C. Furthermore, due to space limit, we conduct additional experiments in Appendix C.4, where we compare different unlearning algorithms' Unlearning Quality across different dataset sizes, unlearning portion parameters, datasets, and model architectures, attacks, and also a linear setting with small privacy budgets to verify our theory.

### 4.1 Experiment settings

We focus on one of the most common tasks in the machine unlearning literature, image classification, and perform experiments on the CIFAR10 dataset [Krizhevsky et al., 2009], which is licensed under CC-BY 4.0. Moreover, we opt for ResNet [He et al., 2016] as the *target model* produced by some learning algorithms LR, whose details can be found in Appendix C.2. Finally, the following is the setup of the unlearning sample inference game $\mathcal{G} = (\mathcal{A}, \mathrm{UL}, \mathcal{D}, \mathbb{P}_\mathcal{D}, \alpha)$ for the evaluation experiment:

- **Initialization**: Since some MIA adversaries require training the so-called *shadow models* using data sampled from the same distribution of the training data used by the target model [Shokri et al., 2017], we start by splitting the whole dataset to accommodate the training of shadow models. In particular, we split the given dataset into two halves, one for training the target model (which we call the *target dataset*), and the other for training shadow models for some MIAs. The target dataset is what we denoted as $\mathcal{D}$ in the game. To initialize the game, we consider a uniform sensitivity distribution $\mathbb{P}_{\mathcal{D}} = \mathcal{U}(\mathcal{D})$ since we do not have any prior preference for the data. The unlearning portion parameter is set to be $\alpha = 0.1$ unless specified. This implies $\mathcal{O}_s(0) = \mathcal{U}(\mathcal{F})$ and $\mathcal{O}_s(1) = \mathcal{U}(\mathcal{T})$, where $s = (\mathcal{R}, \mathcal{F}, \mathcal{T}) \in \mathcal{S}_\alpha$ is the split we choose to use for the game.

- **Challenger Phase**: As mentioned at the beginning of the section, we choose the learning algorithm LR which outputs ResNet as the target model. On the other hand, the corresponding unlearning algorithms UL we select for comparison are: 1.) RETRAIN: retrain from scratch (the gold-standard); 2.) FISHER: Fisher forgetting [Golatkar et al., 2020]; 3.) FTFINAL: fine-tuning final layer [Goel et al., 2023]; 4.) RETRFINAL: retrain final layer [Goel et al., 2023]; 5.) NEGGRAD: negative gradient descent [Golatkar et al., 2020]; 6.) SALUN: saliency unlearning [Fan et al., 2024]; 7.) SSD: selective synaptic dampening [Foster et al., 2024]; 8.) NONE: identity (no unlearning, dummy baseline). Among them, RETRAIN is the gold standard for exact unlearning while NONE is a dummy baseline for reference. All other methods are approximate unlearning methods, with SSD being the most recent state-of-the-art methods.

- **Adversary Phase**: Since we're unaware of any non-weak MIAs, we focus on the following SOTA black-box (weak) MIA adversaries $\mathcal{A}$ to approximate the advantage: 1.) shadow model-based [Shokri et al., 2017]; 2.) correctness-based, confidence-based, modified entropy [Song and Mittal, 2021].

## 4.2 Ablation study

We first provide some empirical evidence that our approximation is reasonable and effective through two lenses: dataset size and $\alpha$.

**Unlearning Quality Versus Dataset Size.** We vary the dataset size to assess how the unlearning quality metric $\mathcal{Q}$ scales with data. Experiments are conducted with $\alpha = 0.1$, and results are shown in Table 1. The relative ranking of unlearning methods remains consistent across sizes, indicating that small-scale evaluations can reliably approximate large-scale performance. This scalability makes $\mathcal{Q}$ an efficient and practical metric for benchmarking unlearning algorithms. The retraining method maintains $\mathcal{Q} \approx 1$, further confirming the theoretical grounding in Theorem 3.3.

Table 1: Unlearning Quality *versus* dataset size $\eta$ (in percentage). The relative ranking of different unlearning methods with added standard deviations.

| UL | $\eta$ | | | |
|---|---|---|---|---|
| | 0.1 | 0.4 | 0.8 | 1.0 |
| RETRFINAL | $0.340 \pm 0.017$ | $0.586 \pm 0.015$ | $0.621 \pm 0.014$ | $0.634 \pm 0.025$ |
| FTFINAL | $0.131 \pm 0.011$ | $0.585 \pm 0.016$ | $0.619 \pm 0.014$ | $0.634 \pm 0.024$ |
| FISHER | $0.751 \pm 0.024$ | $0.679 \pm 0.005$ | $0.734 \pm 0.006$ | $0.791 \pm 0.020$ |
| NEGGRAD | $0.124 \pm 0.010$ | $0.564 \pm 0.018$ | $0.603 \pm 0.014$ | $0.656 \pm 0.035$ |
| SALUN | $0.476 \pm 0.014$ | $0.617 \pm 0.016$ | $0.689 \pm 0.013$ | $0.748 \pm 0.004$ |
| SSD | $0.975 \pm 0.008$ | $0.939 \pm 0.025$ | $0.929 \pm 0.021$ | $0.928 \pm 0.015$ |
| RETRAIN | $0.999 \pm 0.000$ | $0.997 \pm 0.001$ | $0.993 \pm 0.001$ | $0.993 \pm 0.001$ |

**Unlearning Quality Versus $\alpha$.** We further vary the unlearning portion parameter $\alpha$, defined as the ratio of the forget set to the full training set, to investigate how the unlearning quality metric $\mathcal{Q}$ changes. The results are presented in Table 2. We observe that the relative ranking of different unlearning methods remains largely consistent across varying $\alpha$ values. This stability suggests that our metric reliably captures the comparative performance of unlearning algorithms under different data removal proportions.

Table 2: Unlearning Quality *versus* $\alpha$. The relative ranking of different unlearning methods stays mostly consistent across different $\alpha$.

| UL | $\alpha$ | | | |
|---|---|---|---|---|
| | 0.1 | 0.25 | 0.4 | 0.67 |
| RETRFINAL | $0.513 \pm 0.054$ | $0.489 \pm 0.055$ | $0.413 \pm 0.027$ | $0.500 \pm 0.024$ |
| FTFINAL | $0.511 \pm 0.054$ | $0.484 \pm 0.054$ | $0.394 \pm 0.041$ | $0.479 \pm 0.020$ |
| FISHER | $0.904 \pm 0.046$ | $0.871 \pm 0.044$ | $0.908 \pm 0.038$ | $0.810 \pm 0.050$ |
| NEGGRAD | $0.637 \pm 0.105$ | $0.757 \pm 0.073$ | $0.684 \pm 0.016$ | $0.631 \pm 0.088$ |
| SALUN | $0.755 \pm 0.017$ | $0.762 \pm 0.043$ | $0.895 \pm 0.047$ | $0.893 \pm 0.028$ |
| SSD | $0.944 \pm 0.038$ | $0.960 \pm 0.019$ | $0.837 \pm 0.054$ | $0.913 \pm 0.037$ |

## 4.3 Validation of theoretical results

We empirically validate Theorems 3.3 and 3.5. While it is easy to verify **grounding**, i.e., $\mathcal{Q}(\text{RETRAIN}) = 1$, validating the lower-bound of $\mathcal{Q}$ for unlearning algorithms with $(\epsilon, \delta)$-certified removal guarantees is challenging since such algorithms are not known beyond convex models. However, if the model is trained with $(\epsilon, \delta)$-differential privacy (DP) guarantees, then even if we do not apply any unlearning on the model (i.e., UL = NONE), the model still automatically satisfies $(\epsilon, \delta)$-certified removal for any training data point [Guo et al., 2020]. As the DP algorithm exists for non-convex models [Abadi et al., 2016], this suggests that one can analyze the impact of the *DP privacy budget* on $\mathcal{Q}$ of an $(\epsilon, \delta)$-DP model. In particular, we fix $\delta = 10^{-5}$ and consider varying $\epsilon$ to be 50, 150, 600, and $\infty$ ($\infty$ corresponds to no DP training). The corresponding Unlearning Quality results are reported in Table 3.

Firstly, as can be seen from the last row of Table 3, $\mathcal{Q}(\text{RETRAIN}) \approx 1$ with high precision for all $\epsilon$, achieving **grounding** almost perfectly, thus validating Theorem 3.3. Furthermore, Theorem 3.5 suggests that the lower bound of $\mathcal{Q}$ should **negatively correlate** with $\epsilon$. Indeed, empirically, we observe such a trend with high precision, again validating our theoretical findings. Moreover, Table 3 shows that the Unlearning Quality also maintains a **consistent** relative ranking between unlearning algorithms among different $\epsilon$, proving its robustness. Finally, our metric suggests that SSD significantly outperforms

Table 3: Unlearning Quality *versus* DP budgets. We use $^\dagger$ to indicate that the results' standard error of the mean is $< 0.01$ and use $^*$ for $< 0.005$.

| UL | $\epsilon$ | | | |
|---|---|---|---|---|
| | 50 | 150 | 600 | $\infty$ |
| NONE | $0.972^\dagger$ | $0.960^\dagger$ | $0.932^*$ | $0.587^\dagger$ |
| NEGGRAD | $0.980^*$ | $0.975^\dagger$ | $0.953^\dagger$ | $0.628^\dagger$ |
| RETRFINAL | $0.972^\dagger$ | $0.964^\dagger$ | $0.939^\dagger$ | $0.576^\dagger$ |
| FTFINAL | $0.973^*$ | $0.963^\dagger$ | $0.939^\dagger$ | $0.574^\dagger$ |
| FISHER | $0.973^*$ | $0.967^\dagger$ | $0.942^*$ | $0.709^\dagger$ |
| SALUN | $0.979^*$ | $0.972^\dagger$ | $0.945^*$ | $0.689^*$ |
| SSD | $0.996^*$ | $0.988^*$ | $0.981^\dagger$ | $0.888^\dagger$ |
| RETRAIN | $0.998^*$ | $0.996^*$ | $0.997^*$ | $0.993^*$ |

other unlearning methods, which is consistent with the fact that it is the most recent state-of-the-art method among the unlearning methods we evaluate.

**Remark 4.1.** *From Table 3, even for* NONE *with* $\epsilon = \infty$, *the Unlearning Quality* $\mathcal{Q}$ *is still relatively high (0.587). This is partly because the current state-of-the-art (weak) MIA adversaries are not good enough: if the weak adversary becomes better, our evaluation metric can also benefit from this.*

**Remark 4.2.** *We distinguish between A) the computation cost of the proposed unlearning evaluation metric and B) the cost of evaluating the metric itself. The latter is significantly higher due to DP training and is reported in the appendix. In real applications, only the scalable cost of A) is incurred.*

## 4.4 Comparison to other metrics

We compare our Unlearning Quality metric $\mathcal{Q}$ to other existing attack-based evaluation metrics, demonstrating the superiority of the proposed metric. We limit our evaluation to the MIA-based evaluation, and within this category, three MIA-based evaluation metrics are most relevant to our setting [Triantafillou and Kairouz, 2023, Golatkar et al., 2021, Goel et al., 2023]. While none of them enjoy the **grounding** property, in particular, the one proposed by Triantafillou and Kairouz [2023] requires training attacks for every forget data sample, which is extremely time-consuming, and we leave it out from the comparison and focus on comparing our metric with the other two.

The two metrics we will compare are the pure MIA AUC (Area Under Curve) [Golatkar et al., 2021] and the Interclass Confusion (IC) Test [Goel et al., 2023]. The former is straightforward but falls

Table 4: Comparison between IC score and MIA score under different DP budgets. See Table 3 for more context.

(A) IC score *versus* DP budgets.

| $U_L$ | $\epsilon$ | | | |
|---|---|---|---|---|
| | 50 | 150 | 600 | $\infty$ |
| NONE | 0.749$^\dagger$ | 0.720$^\dagger$ | 0.717$^\dagger$ | 0.005$^*$ |
| NEGGRAD | 0.925$^*$ | 0.953$^\dagger$ | 0.946$^\dagger$ | 0.180$^\dagger$ |
| RETRFINAL | 0.931$^*$ | 0.958$^\dagger$ | 0.893$^\dagger$ | 0.033$^\dagger$ |
| FTFINAL | 0.930$^*$ | 0.957$^\dagger$ | 0.893$^\dagger$ | 0.346$^\dagger$ |
| FISHER | 0.743$^*$ | 0.720$^\dagger$ | 0.702$^\dagger$ | 0.344$^\dagger$ |
| SALUN | 0.866$^*$ | 0.887$^*$ | 0.841$^\dagger$ | 0.081$^*$ |
| SSD | 0.976$^*$ | 1.000$^*$ | 0.990$^\dagger$ | 0.174$^\dagger$ |
| RETRAIN | 0.962$^*$ | 0.972$^\dagger$ | 0.976$^\dagger$ | 0.975$^\dagger$ |

(B) MIA score *versus* DP budgets.

| $U_L$ | $\epsilon$ | | | |
|---|---|---|---|---|
| | 50 | 150 | 600 | $\infty$ |
| NONE | 0.451$^\dagger$ | 0.433$^\dagger$ | 0.454$^\dagger$ | 0.380$^\dagger$ |
| NEGGRAD | 0.476$^\dagger$ | 0.482$^\dagger$ | 0.466$^\dagger$ | 0.299$^\dagger$ |
| RETRFINAL | 0.485$^\dagger$ | 0.485$^\dagger$ | 0.472$^\dagger$ | 0.248$^\dagger$ |
| FTFINAL | 0.485$^\dagger$ | 0.485$^\dagger$ | 0.472$^\dagger$ | 0.247$^\dagger$ |
| FISHER | 0.475$^\dagger$ | 0.484$^\dagger$ | 0.463$^\dagger$ | 0.325$^\dagger$ |
| SALUN | 0.488$^*$ | 0.491$^*$ | 0.477$^\dagger$ | 0.268$^*$ |
| SSD | 0.480$^*$ | 0.480$^*$ | 0.468$^*$ | 0.244$^*$ |
| RETRAIN | 0.479$^*$ | 0.491$^*$ | 0.492$^*$ | 0.488$^*$ |

short in many aspects as discussed in the introduction; the latter, on the other hand, is a more refined metric. In brief, the IC test "confuses" a selected set $S$ of two classes by switching their labels and training a model on the modified dataset. It then requests the unlearning algorithm to unlearn $S$ and measures the inter-class error $\gamma \in [0, 1]$ of the unlearned model on $S$. Similar to the advantage, $\gamma$ is the "flipped side" of the unlearning performance, which suggests defining the *IC score* by $1 - \gamma$. Similarly, for the sake of clear comparison, we report the *MIA score* defined as "$1 -$ MIA AUC" as the unlearning performance, where the MIA AUC is calculated on the union of the forget set and test set. We leave the details to Appendix C.3.

We conduct the comparison experiment by again analyzing the relation between the *DP privacy budget* versus the *evaluation result* of an $(\epsilon, \delta)$-DP model for the two metrics we are comparing with under the same setup as in Table 3. Specifically, we let $\delta = 10^{-5}$ and consider varying $\epsilon$ from 50 to $\infty$, and we look into two aspects: 1) *negative correlation* with $\epsilon$; 2) *consistency* w.r.t. $\epsilon$. The results are shown in Table 4. Firstly, we see that according to Table 4(A), the IC test **fails** to produce a *negatively correlated* evaluation result with $\epsilon$. For instance, the IC score for NEGGRAD is notably lower at $\epsilon = 50$ than at $\epsilon = 150$, and RETRFINAL and FTFINAL also demonstrate a higher IC score at $\epsilon = 150$ than at $\epsilon = 600$. Furthermore, in terms of *consistency* w.r.t. $\epsilon$, we again see that the IC test **fails** to satisfy this property, unlike the proposed Unlearning Quality metric $\mathcal{Q}$. For example, while NONE is better than NEGGRAD at $\epsilon = 50$, this relative ranking is not maintained at $\epsilon = \infty$. Such an inconsistency happens multiple times across Table 4(A). A similar story can be told for the MIA AUC, where from Table 4(B), we see that MIA AUC also **fails** to produce a similar trend as $\mathcal{Q}$ where the evaluation results are *negatively correlated* with $\epsilon$. For instance, the MIA scores for NEGGRAD and FISHER are notably higher at $\epsilon = 150$ than at $\epsilon = 600$. Furthermore, in terms of *consistency* w.r.t. $\epsilon$, we see that while NEGGRAD outperforms FISHER at $\epsilon = 50$ and $\epsilon = 600$, it performs worse than FISHER at $\epsilon = 150$, which is **inconsistent**. Overall, both the IC test and the MIA AUC fail to satisfy the properties that we have established for the Unlearning Quality, demonstrating our superiority.

## 5 Conclusion

In this work, we developed a game-theoretical framework named the *unlearning sample inference game* and proposed a novel metric for evaluating the data removal efficacy of approximate unlearning methods. Our approach is rooted in the concept of "advantage," borrowed from cryptography, to quantify the success of an MIA adversary in differentiating forget data from test data given an unlearned model. This metric enjoys zero grounding for the theoretically optimal retraining method, scales with the privacy budget of certified unlearning methods, and can take advantage (as opposed to suffering from conflicts) of various MIA methods, which are desirable properties that existing MIA-based evaluation metrics fail to satisfy. We also propose a practical tool — the *SWAP* test — to efficiently approximate the proposed metric. Our empirical findings reveal that the proposed metric effectively captures the nuances of machine unlearning, demonstrating its robustness across varying dataset sizes and its adaptability to the constraints of differential privacy budgets. The ability to maintain a discernible difference and a partial order among unlearning methods, regardless of dataset size, highlights the practical utility of our approach. By bridging theoretical concepts with empirical analysis, our work lays a solid foundation for reliable empirical evaluation of machine unlearning and paves the way for the development of more effective unlearning algorithms.

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

# A  Additional related work

**Machine Unlearning.**   Techniques like data sharding [Bourtoule et al., 2021, Chen et al., 2022] partition the training process in such a way that only a portion of the model needs to be retrained when removing a subset of the dataset, reducing the computational burden compared to retraining the entire model. For example, Guo et al. [2020], Neel et al. [2021], Chien et al. [2023] analyzed the influence of removed data on linear or convex models and proposed gradient-based updates on model parameters to remove this influence. Chourasia and Shah [2023] proposed an unlearning method that appears similar to ours but differs in key aspects, particularly in the definition of advantage, a term whose meaning varies by threat model. Their threat model leads to a criterion for unlearning effectiveness, but like theory-based approaches such as certified removal, it is hard to empirically evaluate for most approximate methods lacking guarantees. In contrast, our threat model is tailored to enable a practical and novel evaluation metric, addressing a key gap in unlearning research.

**Retraining-based Evaluation.**   Generally, retraining-based evaluation seeks to compare unlearned models to retrained models. As introduced in the works by Golatkar et al. [2021], He et al. [2021], Golatkar et al. [2020], model accuracy on the forget set should be similar to the accuracy on the test set as if the forget set never exists in the training set. Peste et al. [2021] proposed an evaluation metric based on the normalized confusion matrix element-wise difference on selected data samples. Golatkar et al. [2021] proposed using relearn time, which is the additional time to use for unlearned models to perform comparably to retrained models. The authors also proposed to measure the $\ell_1$ distance between the final activations of the scrubbed weights and the retrained model. Wu et al. [2020], Izzo et al. [2021] turned to $\ell_2$ distance of weight parameters between unlearned models and retrained models. In general, beyond the need for additional implementation and the lower computational efficiency inherent in retraining-based evaluations, a more critical issue is the influence of random factors. As discussed by Cretu et al. [2023], such random factors, including the sequence of data batches and the initial configuration of models, can lead to the unaligned storage of information within models. This misalignment may foster implicit biases favoring certain retrained models.

**Theory-based Evaluation.**   Some literature tries to characterize data removal efficacy by requiring a strict theoretical guarantee for the unlearned models. However, these methods have strong model assumptions, such as convexity or linearity, or require inefficient white-box access to target models, thus limiting their applicability in practice. For example, Guo et al. [2020], Neel et al. [2021], Chien et al. [2023] focus on the notion of the certified removal (CR), which requires that the unlearned model cannot be statistically distinguished from the retrained model. By definition, CR is parametrized by privacy parameters called *privacy budgets*, which quantify the level of statistical indistinguishability. Hence, models with CR guarantees will intrinsically satisfy an "evaluation metric" induced by the definition of CR, acting as a form of "evaluation." On the other hand, Becker and Liebig [2022] adopted an information-theoretical perspective and turned to epistemic uncertainty to evaluate the information remaining after unlearning. A concurrent study [Brimhall et al., 2025] that appeared later than this paper proposes a framework for evaluating machine unlearning called computational unlearning, inspired by cryptographic indistinguishability games between retrained models and unlearned models. Their framework shares similar motivations to ours but has different technical assumptions.

**Attack-based Evaluation.**   Since attacks are the most direct way to interpret privacy risks, attack-based evaluation is a common metric in unlearning literature. The classical approach is to directly calculate the MIA accuracy using various kinds of MIAs [Graves et al., 2020, Kurmanji et al., 2023]. One kind of MIA utilizes shadow models [Shokri et al., 2017], which are trained with the same model structure as the original models but on a shadow dataset sampled with the same data sampling distribution. Moreover, some MIAs calculate membership scores based on correctness and confidence [Song and Mittal, 2021]. Some evaluation metrics do move beyond the vanilla MIA accuracy. For example, Triantafillou and Kairouz [2023] leveraged hypothesis testing coupled with MIAs to compute an estimated privacy budget for each unlearning method, which gives a rather rigorous estimation of unlearning efficacy. Hayes et al. [2024] proposed a novel MIA towards machine unlearning based on Likelihood Ratio Attack and evaluated machine unlearning through a combination of the predicted membership probability and the *balanced* MIA accuracy on test and forget sets. They designed a new MIA attack with a similar attack-defense game framework. There are other evaluation metrics also based on MIAs, but with different focuses. However, as they still use MIA accuracy as the evaluation metric, the game itself doesn't bring much for their evaluation framework other than a clear experiment procedure. Goel et al. [2023] proposed an Interclass

Confusion (IC) test that manipulates the input dataset to evaluate both model indistinguishability and property generalization. However, their metric is less direct in terms of interpreting real-life privacy risks. Lastly, For example, Chen et al. [2021] proposed a novel metric based on MIAs that know both learned and unlearned models with a focus on how much information is deleted rather than how much information is left after the unlearning process. Sommer et al. [2020] provided a backdoor verification mechanism for Machine Learning as a Service (MLaaS), which benefits an individual user valuing his/her privacy to verify the efficacy of unlearning. They focus more on user-level verification rather than model-level evaluation.

# B Omitted details from Section 3

## B.1 More details of advantage

In cryptographic games, there are two interacting players: a benign player named *challenger* representing the cryptographic protocol under evaluation (corresponding to the unlearning algorithm in our context), and an *adversary* attempts to compromise the security properties of the challenger. The game proceeds in several phases, including an initialization phase where the game is initialized with specific configuration parameters, a challenger phase where the challenger performs the cryptographic protocol, and an adversary phase where the adversary queries allowed by the game's rules and generates a guess of the secret protected by the challenger. Finally, the game concludes with a win or a loss for the adversary, depending on their guess. In the context of machine unlearning, the goal of the adversary is to guess whether certain given data comes from the set to be unlearned (the forget set) or the set never used in training (the test set), based on access to the unlearned model.

The notion of advantage quantifies, in probabilistic terms, how effectively an adversary can win the game when it is played repeatedly. It is often defined as the difference in the adversary's accepting rate between two distinct scenarios (e.g., with or without access to information potentially leaked by the cryptographic protocol) [Katz and Lindell, 2007]. In the context of machine unlearning, the two scenarios can refer to the data given to the adversary coming from either the forget set or the test set, respectively. The game is constructed such that, if the cryptographic protocol is perfectly secure (i.e., the unlearned model has completely erased the information of the forget set), the adversary's advantage is expected to be zero, making it a well-calibrated measure of the protocol's security.

## B.2 Design choices

In this section, we justify some of our design choices when designing the unlearning sample inference game. Most of them are of practical consideration, while some are for the convenience of analysis.

**Uniform Splitting.** At the initialization phase, we choose the split uniformly rather than allowing sampling from an arbitrary distribution. The reason is two-fold: Firstly, since this sampling strategy corresponds to the so-called *i.i.d. unlearning* setup [Qu et al., 2024], i.e., the unlearned samples will be drawn from a distribution of $\mathcal{D}$ in an i.i.d. fashion. In this regard, uniformly splitting the dataset corresponds to a uniform distribution of $\mathcal{D}$ for the unlearned samples to be drawn from. This is the most commonly used sampling strategy when evaluating unlearning algorithms since it's difficult to estimate the probability that data will be requested to be unlearned.

Secondly, Qu et al. [2024] acknowledged the significantly greater difficulty of non-i.i.d. unlearning compared to i.i.d. unlearning empirically. A classic example of non-i.i.d. unlearning is the process of unlearning an entire class of data, where a subset of data shares the same label. Conversely, even non-uniform splitting complicates the analysis, leading to the breakdown of our theoretical results. Specifically, generalizing both Theorem 3.3 and Theorem 3.5 becomes non-trivial. Overall, non-uniform splitting presents obstacles both empirically and theoretically.

**Intrinsic Learning Algorithm for Challenger.** The challenger, which we denote as UL, *has a learning algorithm* LR *in mind* in our formulation. This is because the existing theory-based unlearning method, such as the certified removal algorithm [Guo et al., 2020] as defined in Definition B.3, is achieved by a combination of the learning algorithm and a corresponding unlearning method to support unlearning request with theoretical guarantees. In other words, given an arbitrary learning algorithm LR, it's unlikely to design an efficient unlearning algorithm UL with strong theoretical guarantees, at least this is not captured by the notion of certified removal. Hence, allowing the challenger to choose its learning algorithm accommodates this situation.

**Black-box Adversary v.s. White-box Adversary.** By default, we assume that $m$ is given to $\mathcal{A}$ in a *black-box* fashion, i.e., $\mathcal{A}$ only has oracle access to $m$. However, our framework can also adapt to white-box adversaries, which requires full model parameters of $m$. The only difference is that the efficiency definition changes accordingly, i.e., polynomial time in the size of $|\mathcal{D}|$ for a black-box adversary or polynomial time in the number of parameters of $m$ for a white-box adversary.

**Strong adversary in practice.** As discussed in Section 3.5, the current state-of-the-art MIA adversaries are all *weak*. While it is possible to formulate the unlearning sample inference game entirely with the weak adversary, we discuss the rationale behind considering the strong adversary. One of the apparent reasons is simply that the strong adversary encompasses the weak adversary, thereby enhancing the generality of our framework and theory. However, we argue that the strong adversary is more practical in many real-world scenarios, and bringing this stronger notion has further practical impacts beyond blindly generalizing our model.

Consider a scenario where an adversary conducts a membership inference attack on a large scale. We argue that in practice, it is more reasonable to aim for a high *overall membership accuracy* of a set of carefully chosen data points, rather than the individual membership status of each of them. For example, consider the case where we are interested in images sourced from the internet. In this case, it is safe to assume that images within the same webpage are either all included in the model training dataset or none are, if we assume that the training data is collected via some reasonable data mining algorithms. In such a case, the ability of an adversary to infer the common membership for a group of data points from a particular webpage becomes desirable as it is likely to enhance the overall MIA accuracy. We believe that this stronger notion of MIA adversary has more practical impacts and reflects the common practice when deploying the membership inference attack; therefore, we choose to formulate the unlearning sample inference game with it.

## B.3   Proof of Theorem 3.3

We now prove Theorem 3.3. We repeat the statement for convenience.

**Theorem B.1.** *For any (potentially inefficient) adversary $\mathcal{A}$, its advantage against the retraining method* RETRAIN *in an unlearning sample inference game* $\mathcal{G} = (\text{RETRAIN}, \mathcal{A}, \mathcal{D}, \mathbb{P}_{\mathcal{D}}, \alpha)$ *is zero, i.e.,* $\text{Adv}(\mathcal{A}, \text{RETRAIN}) = 0$.

*Proof.* Firstly, we may partition the collection of all the possible dataset splits $\mathcal{S}_\alpha$ by fixing the retain sets $\mathcal{R} \subset \mathcal{D}$. Specifically, denote the collection of dataset splits with the retain set to be $\mathcal{R}$ as $\mathcal{S}_\alpha[\mathcal{R}] := \{s \in \mathcal{S}_\alpha : s = (\mathcal{R}, \cdot, \cdot)\}$. With the usual convention, when there's no dataset split corresponds to $\mathcal{R}$, $\mathcal{S}_\alpha[\mathcal{R}] = \varnothing$. Observe that for any $s \in \mathcal{S}_\alpha[\mathcal{R}]$, we can pair it up with another dataset split that *swaps* the forget and test sets in $s$. In other words, for any $s = (\mathcal{R}, \mathcal{F}, \mathcal{T}) \in \mathcal{S}_\alpha[\mathcal{R}]$, we see that $(\mathcal{R}, \mathcal{T}, \mathcal{F})$ is also in $\mathcal{S}_\alpha[\mathcal{R}]$ since we assume $|\mathcal{F}| = |\mathcal{T}|$, every dataset split will be paired. In addition, since $\mathcal{R}$ is fixed in $\mathcal{S}_\alpha[\mathcal{R}]$, we know that $\mathbb{P}_{\mathcal{M}}(\text{RETRAIN}, s) =: \mathbb{P}_{\mathcal{M}}(\mathcal{R})$ is the same for all $s \in \mathcal{S}_\alpha[\mathcal{R}]$ since the unlearning algorithm UL is RETRAIN, i.e., it only depends on $\mathcal{R}$. With these observations, we can then combine the paired dataset splits within the expectation.

Specifically, for any $s \in \mathcal{S}_\alpha[\mathcal{R}]$, let $s' \in \mathcal{S}_\alpha[\mathcal{R}]$ to be $s$'s pair, i.e., if $s = (\mathcal{R}, \mathcal{F}, \mathcal{T})$ for some $\mathcal{F}$ and $\mathcal{T}$, then $s' = (\mathcal{R}, \mathcal{T}, \mathcal{F})$. Finally, for a given $\mathcal{R}$, let's denote the collection of all such pairs as

$$\mathcal{P}_{\mathcal{R}} := \{\{s, s'\} \subset \mathcal{S}_\alpha[\mathcal{R}] : s = (\mathcal{R}, \mathcal{F}, \mathcal{T}), s' = (\mathcal{R}, \mathcal{T}, \mathcal{F}) \text{ for some } \mathcal{F}, \mathcal{T}\},$$

where we use a set rather than an ordered list for the pair $s, s'$ since we do not want to deal with repetitions. Observe that $\mathcal{O}_s(0) = \mathcal{O}_{s'}(1)$ and $\mathcal{O}_s(1) = \mathcal{O}_{s'}(0)$ since the oracles are constructed

with respect to the same preference distribution for all data splits. Hence, we have

$$\mathrm{Adv}(\mathcal{A}, \text{RETRAIN})$$

$$= \frac{1}{|\mathcal{S}_\alpha|} \left| \sum_{s \in \mathcal{S}_\alpha} \left( \Pr_{\substack{m \sim \mathbb{P}_\mathcal{M}(\text{RETRAIN}, s) \\ \mathcal{O} = \mathcal{O}_s(0)}} (\mathcal{A}^\mathcal{O}(m) = 1) - \Pr_{\substack{m \sim \mathbb{P}_\mathcal{M}(\text{RETRAIN}, s) \\ \mathcal{O} = \mathcal{O}_s(1)}} (\mathcal{A}^\mathcal{O}(m) = 1) \right) \right|$$

$$= \frac{1}{|\mathcal{S}_\alpha|} \left| \sum_{\mathcal{R} \subset \mathcal{D}} \sum_{\{s, s'\} \in \mathcal{P}_\mathcal{R}} \left( \Pr_{\substack{m \sim \mathbb{P}_\mathcal{M}(\text{RETRAIN}, s) \\ \mathcal{O} = \mathcal{O}_s(0)}} (\mathcal{A}^\mathcal{O}(m) = 1) - \Pr_{\substack{m \sim \mathbb{P}_\mathcal{M}(\text{RETRAIN}, s) \\ \mathcal{O} = \mathcal{O}_s(1)}} (\mathcal{A}^\mathcal{O}(m) = 1) \right.\right.$$

$$\left.\left. + \Pr_{\substack{m \sim \mathbb{P}_\mathcal{M}(\text{RETRAIN}, s') \\ \mathcal{O} = \mathcal{O}_{s'}(0)}} (\mathcal{A}^\mathcal{O}(m) = 1) - \Pr_{\substack{m \sim \mathbb{P}_\mathcal{M}(\text{RETRAIN}, s') \\ \mathcal{O} = \mathcal{O}_{s'}(1)}} (\mathcal{A}^\mathcal{O}(m) = 1) \right) \right|$$

$$= \frac{1}{|\mathcal{S}_\alpha|} \left| \sum_{\mathcal{R} \subset \mathcal{D}} \sum_{\{s, s'\} \in \mathcal{P}_\mathcal{R}} \left( \Pr_{\substack{m \sim \mathbb{P}_\mathcal{M}(\mathcal{R}) \\ \mathcal{O} = \mathcal{O}_s(0)}} (\mathcal{A}^\mathcal{O}(m) = 1) - \Pr_{\substack{m \sim \mathbb{P}_\mathcal{M}(\mathcal{R}) \\ \mathcal{O} = \mathcal{O}_s(1)}} (\mathcal{A}^\mathcal{O}(m) = 1) \right.\right.$$

$$\left.\left. + \Pr_{\substack{m \sim \mathbb{P}_\mathcal{M}(\mathcal{R}) \\ \mathcal{O} = \mathcal{O}_s(1)}} (\mathcal{A}^\mathcal{O}(m) = 1) - \Pr_{\substack{m \sim \mathbb{P}_\mathcal{M}(\mathcal{R}) \\ \mathcal{O} = \mathcal{O}_s(0)}} (\mathcal{A}^\mathcal{O}(m) = 1) \right) \right| = 0.$$

$\square$

Before we end this section, we remark that the above proof implies that the *SWAP* test also grounds the advantage to zero:

**Remark B.2.** *Consider a pair of swapped splits $s$ and $s'$. Observe that for* RETRAIN, $\mathbb{P}_\mathcal{M}(\text{RETRAIN}, s) = \mathbb{P}_\mathcal{M}(\text{RETRAIN}, s') =: \mathbb{P}_\mathcal{M}(\mathcal{R})$ *since this probability only depends on $\mathcal{R}$, which is the same for $s$ and $s'$. With $\mathcal{O}_s(0) = \mathcal{O}_{s'}(1)$ and $\mathcal{O}_s(1) = \mathcal{O}_{s'}(0)$, we have*

$$\overline{\mathrm{Adv}}_{\{s, s'\}}(\mathcal{A}, \text{RETRAIN}) = \frac{1}{2} \left| \Pr_{m \sim \mathbb{P}_\mathcal{M}(\mathcal{R})} (\mathcal{A}^{\mathcal{O}_s(0)}(m) = 1) - \Pr_{m \sim \mathbb{P}_\mathcal{M}(\mathcal{R})} (\mathcal{A}^{\mathcal{O}_s(1)}(m) = 1) \right.$$

$$\left. + \Pr_{m \sim \mathbb{P}_\mathcal{M}(\mathcal{R})} (\mathcal{A}^{\mathcal{O}_s(1)}(m) = 1) - \Pr_{m \sim \mathbb{P}_\mathcal{M}(\mathcal{R})} (\mathcal{A}^{\mathcal{O}_s(0)}(m) = 1) \right| = 0.$$

## B.4 Proof of Theorem 3.5

We prove Theorem 3.5 in this section. Before this, we formally introduce the notion of *certified removal*.

**Definition B.3** (Certified Removal [Guo et al., 2020]). *For a fixed dataset $\mathcal{D}$, let* LR *and* UL *be a learning and an unlearning algorithm respectively, and denote $\mathcal{H} := \mathrm{im}(\text{LR}) \cup \mathrm{im}(\text{UL})$[6] to be the hypothesis class containing all possible models that can be produced by* LR *and* UL. *Then, for any $\epsilon, \delta > 0$, the unlearning algorithm* UL *is said to be $(\epsilon, \delta)$-certified removal if for any $\mathcal{W} \subset \mathcal{H}$ and for any disjoint $\mathcal{R}, \mathcal{F} \subseteq \mathcal{D}$ (do not need to satisfy restriction (a) and (b)),*

$$\Pr(\text{UL}(\text{LR}(\mathcal{R} \cup \mathcal{F}), \mathcal{F}) \in \mathcal{W}) \leq e^\epsilon \Pr(\text{RETRAIN}(\text{LR}(\mathcal{R} \cup \mathcal{F}), \mathcal{F}) \in \mathcal{W}) + \delta;$$
$$\Pr(\text{RETRAIN}(\text{LR}(\mathcal{R} \cup \mathcal{F}), \mathcal{F}) \in \mathcal{W}) \leq e^\epsilon \Pr(\text{UL}(\text{LR}(\mathcal{R} \cup \mathcal{F}), \mathcal{F}) \in \mathcal{W}) + \delta.$$

We note that as $\text{RETRAIN}(\text{LR}(\mathcal{R} \cup \mathcal{F}), \mathcal{F}) = \text{LR}(\mathcal{R})$, the above can be simplified to

$$\Pr(\text{UL}(\text{LR}(\mathcal{R} \cup \mathcal{F}), \mathcal{F}) \in \mathcal{W}) \leq e^\epsilon \Pr(\text{LR}(\mathcal{R}) \in \mathcal{W}) + \delta$$
$$\Pr(\text{LR}(\mathcal{R}) \in \mathcal{W}) \leq e^\epsilon \Pr(\text{UL}(\text{LR}(\mathcal{R} \cup \mathcal{F}), \mathcal{F}) \in \mathcal{W}) + \delta.$$

Now, we restate Theorem 3.5 for convenience.

---

[6]Here, $\mathrm{im}(\cdot)$ denote the image of a function.

**Theorem B.4.** *Given an $(\epsilon, \delta)$-certified removal unlearning algorithm* UL *with some $\epsilon, \delta > 0$, for any (potentially inefficient) adversary $\mathcal{A}$ against* UL *in an unlearning sample inference game $\mathcal{G}$, we have*

$$\text{Adv}(\mathcal{A}, \text{UL}) \leq 2 \cdot \left(1 - \frac{2 - 2\delta}{e^\epsilon + 1}\right).$$

*Proof.* We start by considering an attack as differentiating between the following two hypotheses: the unlearning and the retraining. In particular, given a specific dataset split $s = (\mathcal{R}, \mathcal{F}, \mathcal{T}) \in \mathcal{S}_\alpha$ and an model $m$, consider

$$H_1 : m = \text{UL}(\text{LR}(\mathcal{R} \cup \mathcal{F}), \mathcal{F}), \text{ and } H_2 : m = \text{LR}(\mathcal{R}) = \text{RETRAIN}(\text{LR}(\mathcal{R} \cup \mathcal{F}), \mathcal{F}).$$

Alternatively, by writing the distribution of the unlearned models and the retrained models as $\mathbb{P}_{\mathcal{M}}(\text{UL}, s)$ and $\mathbb{P}_{\mathcal{M}}(\text{RETRAIN}, s)$, respectively, we may instead write

$$H_1 : m \sim \mathbb{P}_{\mathcal{M}}(\text{UL}, s), \text{ and } H_2 : m \sim \mathbb{P}_{\mathcal{M}}(\text{RETRAIN}, s).$$

It turns out that by looking at the *type-I error $\alpha$* and *type-II error $\beta$*, we can control the advantage of the adversary in this game easily. Firstly, denote the model produced under $H_1$ as $m_1$, then under $H_1$, the accuracy of the adversary is $\Pr_{m_1 \sim \mathbb{P}_{\mathcal{M}}(\text{UL},s)}(\mathcal{A}(m_1) = 1) = 1 - \alpha$. Similarly, by denoting the model produced under $H_2$ as $m_2$, we have $\Pr_{m_2 \sim \mathbb{P}_{\mathcal{M}}(\text{RETRAIN},s)}(\mathcal{A}(m_2) = 1) = \beta$. Therefore, for this specific dataset split $s$, let's define the advantage of this adversary $\mathcal{A}$ for this attack as[7]

$$\widehat{\text{Adv}}_s(\mathcal{A}) := |1 - \alpha - \beta|.$$

The upshot is that since UL is an $(\epsilon, \delta)$-certified removal unlearning algorithm (Definition B.3), it is possible to control $\alpha$ and $\beta$, hence $\widehat{\text{Adv}}_s(\mathcal{A})$. To achieve this, since from the definition of certified removal, we're dealing with sub-collections of models, it helps to write $\alpha$ and $\beta$ differently.

Let $\mathcal{H} := \text{supp}(\mathbb{P}_{\mathcal{M}}(\text{UL}, s)) \cup \text{supp}(\mathbb{P}_{\mathcal{M}}(\text{RETRAIN}, s))$ be the collection of all possible models, and denote $\mathcal{B} \subseteq \mathcal{H}$ to be the collection of models that the adversary $\mathcal{A}$ accepts, and $\mathcal{B}^c$ to denote its complement, i.e., the collection of models that the adversary $\mathcal{A}$ rejects. We can then re-write the type-I and type-II errors as

- Type-I error $\alpha$: probability of rejecting $H_1$ when $H_1$ is true, i.e., $\alpha = \Pr(m_1 \in \mathcal{B}^c \mid s) = 1 - \Pr(m_1 \in \mathcal{B} \mid s)$.

- Type-II error $\beta$: probability of accepting $H_2$ when $H_2$ is false, i.e., $\beta = \Pr(m_2 \in \mathcal{B} \mid s)$.

With this interpretation and the fact that UL is $(\epsilon, \delta)$-certified removal, we know that

- $\Pr(m_1 \in \mathcal{B} \mid s) \leq e^\epsilon \Pr(m_2 \in \mathcal{B} \mid s) + \delta$, and

- $\Pr(m_2 \in \mathcal{B}^c \mid s) \leq e^\epsilon \Pr(m_1 \in \mathcal{B}^c \mid s) + \delta$.

Combining the above, we have $1 - \alpha \leq e^\epsilon \beta + \delta$ and $1 - \beta \leq e^\epsilon \alpha + \delta$. Hence,

$$\beta \geq \max\{0, 1 - \delta - e^\epsilon \alpha, e^{-\epsilon}(1 - \delta - \alpha)\}.$$

We then seek to get the minimum of $\alpha + \beta$, we have

$$\alpha + \beta \geq \max\{\alpha, 1 - \delta - e^\epsilon \alpha + \alpha, e^{-\epsilon}(1 - \delta - \alpha) + \alpha\}.$$

To get a lower bound, consider the minimum among the last two, i.e., consider solving $\alpha$ when $1 - \delta - e^\epsilon \alpha + \alpha = e^{-\epsilon}(1 - \delta - \alpha) + \alpha$, leading to

$$(e^{-\epsilon} - e^\epsilon)\alpha = e^{-\epsilon}(1 - \delta) - (1 - \delta) = (e^{-\epsilon} - 1)(1 - \delta) \implies \alpha = \frac{(e^{-\epsilon} - 1)(1 - \delta)}{e^{-\epsilon} - e^\epsilon}.$$

Hence, we have

$$\alpha + \beta \geq 1 - \delta + \alpha(1 - e^\epsilon) = 1 - \delta + \frac{(e^{-\epsilon} - 1)(1 - \delta)}{e^{-\epsilon} - e^\epsilon}(1 - e^\epsilon) = (1 - \delta)\frac{2e^{-\epsilon} - 2}{e^{-\epsilon} - e^\epsilon} = (1 - \delta)\frac{2 - 2e^\epsilon}{1 - e^{2\epsilon}},$$

---

[7]Note that this is different from the advantage we defined before since the attack is different, hence we use a different notation.

with the elementary identity $1 - e^{2\epsilon} = (1 + e^\epsilon)(1 - e^\epsilon)$, we finally get

$$\alpha + \beta \geq \frac{2 - 2\delta}{e^\epsilon + 1}.$$

On the other hand, considering the "dual attack" that predicts the opposite as the original attack, that is, we flip $\mathcal{B}$ and $\mathcal{B}^c$. In this case, the type-I error and the type-II error become $\alpha$ and $1 - \beta$, respectively. Following the same procedures, we'll have $\alpha + \beta \leq 2 - \frac{2-2\delta}{e^\epsilon+1}$.

Note that the definition of $(\epsilon, \delta)$-certified removal is independent of the dataset split $s$, hence, the above derivation works for all $s$. In particular, the advantage of any adversary differentiating $H_1$ and $H_2$ for any $s$ is upper bounded by

$$\widehat{\mathrm{Adv}}_s(\mathcal{A}) = |1 - \alpha - \beta| \leq 1 - \frac{2 - 2\delta}{e^\epsilon + 1} =: \tau,$$

where we denote $\tau$ to be the upper bound of advantage. This means that **any** adversaries trying to differentiate between retrain models and certified unlearned models are upper bounded in terms of their advantage, and an explicit upper bound is given by $\tau$.

We now show a reduction from the unlearning sample inference game to the above. Firstly, we construct two attacks based on the adversary $\mathcal{A}$ in the unlearning sample inference game, which tries to differentiate between the data point $x$ is sampled from the forget set $\mathcal{F}$ or the test set $\mathcal{T}$. This can be formulated through the following hypothesis testing:

$$H_3 \colon x \in \mathcal{F}, \text{ and } H_4 \colon x \in \mathcal{T}.$$

In this viewpoint, the unlearning sample inference game can be thought of as deciding between $H_3$ and $H_4$. Since the upper bound we get for differentiating between $H_1$ and $H_2$ holds for any efficient adversaries, therefore, we can construct an attack for deciding between $H_1$ and $H_2$ using adversaries for deciding between $H_3$ and $H_4$. This allows us to upper bound the advantage for the latter adversaries.

Given any adversaries $\mathcal{A}$ for differentiating $H_3$ and $H_4$, i.e., any adversaries in the unlearning sample inference game, we start by constructing our first adversary $\mathcal{A}_1$ for differentiating $H_1$ and $H_2$ as follows:

- In the left world ($H_1$), feed the certified unlearned model to $\mathcal{A}$; in the right world $H_2$, feed the retrained model to $\mathcal{A}$.

- We create a random oracle $\mathcal{O}_s(0)$ for $\mathcal{A}$, i.e., we let the adversary $\mathcal{A}$ decide on $\mathcal{F}$. We then let $\mathcal{A}_1$ output as $\mathcal{A}$.

We note that $\mathcal{A}$ is deciding on $\mathcal{F}$, the advantage of $\mathcal{A}_1$ is

$$\widehat{\mathrm{Adv}}_s(\mathcal{A}_1) = \left| \Pr_{\substack{m_1 \sim \mathbb{P}_{\mathcal{M}}(\mathrm{UL}, s) \\ \mathcal{O}=\mathcal{O}_s(0)}} (\mathcal{A}^{\mathcal{O}}(m_1) = 1) - \Pr_{\substack{m_2 \sim \mathbb{P}_{\mathcal{M}}(\mathrm{RETRAIN}, s) \\ \mathcal{O}=\mathcal{O}_s(0)}} (\mathcal{A}^{\mathcal{O}}(m_2) = 1) \right|.$$

We can also induce the average of the advantage over all dataset splits is upper bounded by the maximal advantage taken over all dataset splits:

$$\frac{1}{|\mathcal{S}_\alpha|} \left| \sum_{s \in \mathcal{S}_\alpha} \Pr_{\substack{m_1 \sim \mathbb{P}_{\mathcal{M}}(\mathrm{UL}, s) \\ \mathcal{O}=\mathcal{O}_s(0)}} (\mathcal{A}^{\mathcal{O}}(m_1) = 1) - \sum_{s \in \mathcal{S}_\alpha} \Pr_{\substack{m_2 \sim \mathbb{P}_{\mathcal{M}}(\mathrm{RETRAIN}, s) \\ \mathcal{O}=\mathcal{O}_s(0)}} (\mathcal{A}^{\mathcal{O}}(m_2) = 1) \right|$$

$$\leq \frac{1}{|\mathcal{S}_\alpha|} \sum_{s \in \mathcal{S}_\alpha} \left| \Pr_{\substack{m_1 \sim \mathbb{P}_{\mathcal{M}}(\mathrm{UL}, s) \\ \mathcal{O}=\mathcal{O}_s(0)}} (\mathcal{A}^{\mathcal{O}}(m_1) = 1) - \Pr_{\substack{m_2 \sim \mathbb{P}_{\mathcal{M}}(\mathrm{RETRAIN}, s) \\ \mathcal{O}=\mathcal{O}_s(0)}} (\mathcal{A}^{\mathcal{O}}(m_2) = 1) \right|$$

$$\leq \max_{s \in \mathcal{S}_\alpha} \widehat{\mathrm{Adv}}_s(\mathcal{A}_1).$$

Similarly, we can construct a second adversary $\mathcal{A}_2$ for differentiating $H_1$ and $H_2$ as follows:

- In the left world ($H_1$), feed the certified unlearned model to $\mathcal{A}$. In the right world ($H_2$), feed retrained model to $\mathcal{A}$.

- We create a random oracle $O_s(1)$ for $\mathcal{A}$, i.e., we let the adversary $\mathcal{A}$ decide on $\mathcal{T}$. We then let $\mathcal{A}_2$ outputs as the $\mathcal{A}$.

Since $\mathcal{A}$ is deciding on $\mathcal{T}$, the advantage of $\mathcal{A}_2$ is

$$\widehat{\mathrm{Adv}}_s(\mathcal{A}_2) = \left| \Pr_{\substack{m_1 \sim \mathbb{P}_{\mathcal{M}}(\mathrm{UL},s) \\ \mathcal{O}=\mathcal{O}_s(1)}} (\mathbb{1}(\mathcal{A}^{\mathcal{O}}(m_1)=1)) - \Pr_{\substack{m_2 \sim \mathbb{P}_{\mathcal{M}}(\mathrm{RETRAIN},s) \\ \mathcal{O}=\mathcal{O}_s(1)}} (\mathcal{A}^{\mathcal{O}}(m_2)=1) \right|.$$

Similar to the previous calculation for $\mathcal{A}_1$, the average of the advantage is also upper bounded by the maximal advantage,

$$\frac{1}{|\mathcal{S}_\alpha|} \left| \sum_{s \in \mathcal{S}_\alpha} \Pr_{\substack{m_1 \sim \mathbb{P}_{\mathcal{M}}(\mathrm{UL},s) \\ \mathcal{O}=\mathcal{O}_s(1)}} (\mathcal{A}^{\mathcal{O}}(m_1)=1) - \sum_{s \in \mathcal{S}_\alpha} \Pr_{\substack{m_2 \sim \mathbb{P}_{\mathcal{M}}(\mathrm{RETRAIN},s) \\ \mathcal{O}=\mathcal{O}_s(1)}} (\mathcal{A}^{\mathcal{O}}(m_2)=1) \right|$$

$$\leq \frac{1}{|\mathcal{S}_\alpha|} \sum_{s \in \mathcal{S}_\alpha} \left| \Pr_{\substack{m_1 \sim \mathbb{P}(\mathcal{M}_1,s) \\ \mathcal{O}=\mathcal{O}_s(1)}} (\mathcal{A}^{\mathcal{O}}(m_1)=1) - \Pr_{\substack{m_2 \sim \mathbb{P}_{\mathcal{M}}(\mathrm{RETRAIN},s) \\ \mathcal{O}=\mathcal{O}_s(1)}} (\mathcal{A}^{\mathcal{O}}(m_2)=1) \right|$$

$$\leq \max_{s \in \mathcal{S}_\alpha} \widehat{\mathrm{Adv}}_s(\mathcal{A}_2).$$

Given the above calculation, we can now bound the advantage of $\mathcal{A}$. Firstly, let UL be any certified unlearning method. Then the advantage $\mathrm{Adv}(\mathcal{A}, \mathrm{UL})$ of $\mathcal{A}$ in the unlearning sample inference game (i.e., differentiating between $H_3$ and $H_4$) against UL is

$$\frac{1}{|\mathcal{S}_\alpha|} \left| \sum_{s \in \mathcal{S}_\alpha} \Pr_{\substack{m_1 \sim \mathbb{P}_{\mathcal{M}}(\mathrm{UL},s) \\ \mathcal{O}=\mathcal{O}_s(0)}} (\mathcal{A}^{\mathcal{O}}(m_1)=1) - \sum_{s \in \mathcal{S}_\alpha} \Pr_{\substack{m_1 \sim \mathbb{P}_{\mathcal{M}}(\mathrm{UL},s) \\ \mathcal{O}=\mathcal{O}_s(1)}} (\mathcal{A}^{\mathcal{O}}(m_1)=1) \right|.$$

On the other hand, the advantage $\mathrm{Adv}(\mathcal{A}, \mathrm{RETRAIN})$ of $\mathcal{A}$ against the retraining method RETRAIN can be written as

$$\frac{1}{|\mathcal{S}_\alpha|} \left| \sum_{s \in \mathcal{S}_\alpha} \Pr_{\substack{m_2 \sim \mathbb{P}_{\mathcal{M}}(\mathrm{RETRAIN},s) \\ \mathcal{O}=\mathcal{O}_s(0)}} (\mathcal{A}^{\mathcal{O}}(m_2)=1) - \sum_{s \in \mathcal{S}_\alpha} \Pr_{\substack{m_2 \sim \mathbb{P}_{\mathcal{M}}(\mathrm{RETRAIN},s) \\ \mathcal{O}=\mathcal{O}_s(1)}} (\mathcal{A}^{\mathcal{O}}(m_2)=1) \right|,$$

which is indeed 0 from Theorem 3.3. Combine this with the calculations above, from the reverse triangle inequality,

$$\mathrm{Adv}(\mathcal{A}, \mathrm{UL})$$

$$= \left| \mathrm{Adv}(\mathcal{A}, \mathrm{UL}) - \mathrm{Adv}(\mathcal{A}, \textsc{Retrain}) \right|$$

$$\leq \frac{1}{|\mathcal{S}_\alpha|} \left| \sum_{s \in \mathcal{S}_\alpha} \Pr_{\substack{m_2 \sim \mathbb{P}_{\mathcal{M}}(\textsc{Retrain}, s) \\ \mathcal{O} = \mathcal{O}_s(0)}} (\mathcal{A}^{\mathcal{O}}(m_2) = 1) - \sum_{s \in \mathcal{S}_\alpha} \Pr_{\substack{m_2 \sim \mathbb{P}_{\mathcal{M}}(\textsc{Retrain}, s) \\ \mathcal{O} = \mathcal{O}_s(1)}} (\mathcal{A}^{\mathcal{O}}(m_2) = 1) \right.$$

$$\left. - \sum_{s \in \mathcal{S}_\alpha} \Pr_{\substack{m_1 \sim \mathbb{P}_{\mathcal{M}}(\mathrm{UL}, s) \\ \mathcal{O} = \mathcal{O}_s(0)}} (\mathcal{A}^{\mathcal{O}}(m_1) = 1) + \sum_{s \in \mathcal{S}_\alpha} \Pr_{\substack{m_1 \sim \mathbb{P}_{\mathcal{M}}(\mathrm{UL}, s) \\ \mathcal{O} = \mathcal{O}_s(1)}} (\mathcal{A}^{\mathcal{O}}(m_1) = 1) \right|$$

$$\leq \frac{1}{|\mathcal{S}_\alpha|} \sum_{s \in \mathcal{S}_\alpha} \left| \Pr_{\substack{m_1 \sim \mathbb{P}_{\mathcal{M}}(\mathrm{UL}, s) \\ \mathcal{O} = \mathcal{O}_s(0)}} (\mathcal{A}^{\mathcal{O}}(m_1) = 1) - \Pr_{\substack{m_2 \sim \mathbb{P}_{\mathcal{M}}(\textsc{Retrain}, s) \\ \mathcal{O} = \mathcal{O}_s(0)}} (\mathcal{A}^{\mathcal{O}}(m_2) = 1) \right|$$

$$+ \frac{1}{|\mathcal{S}_\alpha|} \sum_{s \in \mathcal{S}_\alpha} \left| \Pr_{\substack{m_1 \sim \mathbb{P}_{\mathcal{M}}(\mathrm{UL}, s) \\ \mathcal{O} = \mathcal{O}_s(1)}} (\mathcal{A}^{\mathcal{O}}(m_1) = 1 - \Pr_{\substack{m_2 \sim \mathbb{P}_{\mathcal{M}}(\textsc{Retrain}, s) \\ \mathcal{O} = \mathcal{O}_s(1)}} (\mathcal{A}^{\mathcal{O}}(m_2) = 1) \right|$$

$$\leq \max_{s \in \mathcal{S}_\alpha} \widehat{\mathrm{Adv}}_s(\mathcal{A}_1) + \max_{s \in \mathcal{S}_\alpha} \widehat{\mathrm{Adv}}_s(\mathcal{A}_2)$$

$$\leq 2\tau,$$

i.e., the advantage of any adversary against any certified unlearning method is bounded by $2\tau$. $\square$

### B.5 Proof of Proposition 3.7

We now prove Proposition 3.7. We repeat the statement for convenience.

**Proposition B.5.** *For any two dataset splits $s_1, s_2 \in \mathcal{S}_\alpha$ satisfying non-degeneracy assumption, i.e., both $\mathbb{P}_{\mathcal{D}}|_{\mathcal{F}_i}(\mathcal{F}_1 \cap \mathcal{F}_2)$ and $\mathbb{P}_{\mathcal{D}}|_{\mathcal{T}_i}(\mathcal{T}_1 \cap \mathcal{T}_2)$ do not vanish polynomially faster in $|\mathcal{D}|$, then there exists a deterministic and efficient adversary $\mathcal{A}$ such that $\overline{\mathrm{Adv}}_{\{s_1, s_2\}}(\mathcal{A}, \mathrm{UL}) = 1$ for any unlearning method $\mathrm{UL}$. In particular, $\overline{\mathrm{Adv}}_{\{s_1, s_2\}}(\mathcal{A}, \textsc{Retrain}) = 1$.*

*Proof.* Consider any unlearning method $\mathrm{UL}$, and design a random oracle $\mathcal{O}_{s_i}$ based on the split $s_i$ for $i = 1, 2$ and a sensitivity distribution $\mathbb{P}_{\mathcal{D}}$ (which for simplicity, assume to have full support across $\mathcal{D}$), we see that

$$\overline{\mathrm{Adv}}_{\{s_1, s_2\}}(\mathcal{A}, \mathrm{UL}) = \frac{1}{2} \left| \Pr_{\substack{m \sim \mathbb{P}_{\mathcal{M}}(\mathrm{UL}, s_1) \\ \mathcal{O} = \mathcal{O}_{s_1}(0)}} (\mathcal{A}^{\mathcal{O}}(m) = 1) - \Pr_{\substack{m \sim \mathbb{P}_{\mathcal{M}}(\mathrm{UL}, s_1) \\ \mathcal{O} = \mathcal{O}_{s_1}(1)}} (\mathcal{A}^{\mathcal{O}}(m) = 1) \right.$$

$$\left. + \Pr_{\substack{m \sim \mathbb{P}_{\mathcal{M}}(\mathrm{UL}, s_2) \\ \mathcal{O} = \mathcal{O}_{s_2}(0)}} (\mathcal{A}^{\mathcal{O}}(m) = 1) - \Pr_{\substack{m \sim \mathbb{P}_{\mathcal{M}}(\mathrm{UL}, s_2) \\ \mathcal{O} = \mathcal{O}_{s_2}(1)}} (\mathcal{A}^{\mathcal{O}}(m) = 1) \right|.$$

Consider a hard-coded adversary $\mathcal{A}$ which has a look-up table $T$, defined as

$$T(x) = \begin{cases} 1, & \text{if } x \in \mathcal{T}_1 \cap \mathcal{T}_2; \\ 0, & \text{if } x \in \mathcal{F}_1 \cap \mathcal{F}_2; \\ \bot, & \text{otherwise}, \end{cases}$$

where we use $\bot$ to denote an undefined output. Then, $\mathcal{A}$ predicts the bit $b$ used by the oracle as follows: We see that Algorithm 1 with a hard-coded look-up table $T$ has several properties:

- Since it neglects $m$ entirely,
$$\Pr_{\substack{m \sim \mathbb{P}_{\mathcal{M}}(\mathrm{UL}, s_i) \\ \mathcal{O} = \mathcal{O}_{s_i}(b)}} (\mathcal{A}^{\mathcal{O}}(m) = 1) = \Pr_{\mathcal{O} = \mathcal{O}_{s_i}(b)} (\mathcal{A}^{\mathcal{O}} = 1)$$

**Algorithm 1:** Dummy adversary $\mathcal{A}$ against a random 2-sets evaluation

---

**Data:** An unlearned model $m$, a random oracle $\mathcal{O}$
**Result:** A one bit prediction $b$

$b \leftarrow \perp$
**while** $b = \perp$ **do**
$\quad \mid \quad x \sim \mathcal{O}$
$\quad \lfloor \quad b \leftarrow T(x)$
**return** $b$

---

- Under the non-degeneracy assumptions, $\mathcal{A}$ will terminate in polynomial time in $|\mathcal{D}|$. This is because the expected terminating time is inversely proportional to $\mathbb{P}_{\mathcal{D}}|_{\mathcal{F}_i}(\mathcal{F}_1 \cap \mathcal{F}_2)$ and $\mathbb{P}_{\mathcal{D}}|_{\mathcal{T}_i}(\mathcal{T}_1 \cap \mathcal{T}_2)$, hence if these two probabilities does not vanish polynomially faster in $|\mathcal{D}|$ (i.e., the non-degeneracy assumption), then it'll terminate in polynomial time in $|\mathcal{D}|$.

- Whenever $\mathcal{A}$ terminates and outputs an answer, it will be correct, i.e.,

$$\overline{\mathrm{Adv}}_{\{s_1, s_2\}}(\mathcal{A}, \mathrm{UL}) = 1.$$

Since the above argument works for every UL, hence even for the retraining method RETRAIN, we will have $\overline{\mathrm{Adv}}_{\{s_1, s_2\}}(\mathcal{A}, \mathrm{RETRAIN}) = 1$. Intuitively, such a pathological case can happen since there exists some $\mathcal{A}$ which interpolates the "correct answer" for a few splits. Though adversaries may not have access to specific dataset splits, learning-based attacks could still undesirably learn towards this scenario if evaluated only on a few splits. Thus, we should penalize hard-coded adversaries in evaluation. $\qquad \square$

### B.6 Discussion on a naive strategy offsetting MIA accuracy

In this section, we consider a toy example to illustrate the difference between the proposed advantage-based metric and a naive strategy that simply offsets the MIA accuracy for RETRAIN to zero.

Let us assume there are 6 data points $\{A, B, C, D, E, F\}$, and we equally split them into the forget set $\mathcal{F}$ and the test set $\mathcal{T}$. Running MIA against RETRAIN on a retrained model $m^*$ trained on the retain set $\mathcal{R}$ independent of the above 6 data points, the MIA predicts the probability that each data point belongs to the forget set is:

$$\mathrm{MIA}(m^*, A) = 0.7, \quad \mathrm{MIA}(m^*, B) = 0.4, \quad \mathrm{MIA}(m^*, C) = 0.3,$$
$$\mathrm{MIA}(m^*, D) = 0.1, \quad \mathrm{MIA}(m^*, E) = 0.6, \quad \mathrm{MIA}(m^*, F) = 0.8.$$

Assume that the MIA adversary $\mathcal{A}$ chooses the cutoff of predicting a data point belonging to the forget set to be $\geq 0.5$, then the prediction will be

$$\mathcal{A}(A) = 1, \quad \mathcal{A}(B) = 0, \quad \mathcal{A}(C) = 0, \quad \mathcal{A}(D) = 0, \quad \mathcal{A}(E) = 1, \quad \mathcal{A}(F) = 1,$$

where 1 refers to the forget set while 0 refers to the test set. Since the retrained model will be the same regardless of the split of the forget set and the test set, the MIA prediction will be the same regardless of the split as well.

Assuming that we have two imperfect unlearning algorithms, $\mathrm{UL}_1$ and $\mathrm{UL}_2$. For simplicity, we could assume running MIA on the unlearned model $m_1$ by $\mathrm{UL}_1$ will increase the predicted probability on the forget set by 0.1, while that (denoted as $m_2$) by $\mathrm{UL}_2$ will increase it by 0.2. For example, if we set $\mathcal{F} = \{A, B, C\}$ while $\mathcal{T} = \{D, E, F\}$, then the MIA on $\mathrm{UL}_1$ will predict the probability as

$$\mathrm{MIA}(m_1, A) = 0.8, \quad \mathrm{MIA}(m_1, B) = 0.5, \quad \mathrm{MIA}(m_1, C) = 0.4,$$
$$\mathrm{MIA}(m_1, D) = 0.1, \quad \mathrm{MIA}(m_1, E) = 0.6, \quad \mathrm{MIA}(m_1, F) = 0.8,$$

while the MIA on $\mathrm{UL}_2$ will predict the probability as

$$\mathrm{MIA}(m_2, A) = 0.9, \quad \mathrm{MIA}(m_2, B) = 0.6, \quad \mathrm{MIA}(m_2, C) = 0.5,$$
$$\mathrm{MIA}(m_2, D) = 0.1, \quad \mathrm{MIA}(m_2, E) = 0.6, \quad \mathrm{MIA}(m_2, F) = 0.8,$$

Intuitively, $\mathrm{UL}_2$ is worse than $\mathrm{UL}_1$ in terms of unlearning quality. In this simplified setup, we claim the following.

**Claim B.6.** *The advantage calculated by one SWAP test over* RETRAIN *is always* 0, *while the advantage calculated by one SWAP test over* $\mathrm{UL}_1$ *and* $\mathrm{UL}_2$ *are respectively* $1/6$ *and* $1/3$, *all regardless of the exact split of the data points. Hence, the Unlearning Quality for* RETRAIN, $\mathrm{UL}_1$, *and* $\mathrm{UL}_2$ *are respectively* 1, $5/6$, *and* $2/3$, *faithfully reflecting the unlearning algorithms' performances.*

*On the other hand, the "offset MIA accuracy" is dependent on the split of the data. Specifically, when we assign* $\{D, E, F\}$ *as the forget set and* $\{A, B, C\}$ *as the test set, the MIA accuracies for all three methods are the same, making the "offset MIAs" all equal to* 0.5, *failing to capture the unlearning quality.*

*Proof.* Given a split $s$, denote the predicted MIA result as $\hat{Y}_i^s \in \{0, 1\}$, and the actual membership as $Y_i^s = \mathbb{1}_{i \in \mathcal{F}}$ for $i \in \{A, B, \ldots, F\}$. Then, consider a simple adversary $\mathcal{A}$: after getting the predicted MIA probability, i.e., $\Pr(\hat{Y}_i^s = 1)$, we update $\hat{Y}_i^s$ as 1 if $\Pr(\hat{Y}_i^s = 1) \geq 1/2$, and 0 otherwise. Then, the advantage of a particular split $s$ is $\Pr_m(\mathcal{A}^{\mathcal{O}_s(0)}(m) = 1) - \Pr_m(\mathcal{A}^{\mathcal{O}_s(1)}(m) = 1) = \Pr_i(\hat{Y}_i^s = 1 \mid Y_i^s = 0) - \Pr_i(\hat{Y}_i^s = 1 \mid Y_i^s = 1)$ for $i \in \{A, B, \ldots, F\}$. These probabilities are essentially an average of indicator variables: for example, $\Pr_i(\hat{Y}_i^s = 1 \mid Y_i^s = 0) = \sum_{i: Y_i^s = 0} \mathbb{1}_{\hat{Y}_i^s = 1} / |\{i: Y_i^s = 0\}|$, and since we're considering equal-size splits, $|\{i: Y_i^s = 0\}| = |\{i: Y_i^s = 1\}| = 6/2 = 3$ in our example.

To see how the calculation works out, we note that for a SWAP split $(s, s')$, each $i$ appears in the forget set and the test set exactly once when calculating the SWAP advantage, i.e., $Y_i^s = 1$ and $Y_i^{s'} = 0$ or $Y_i^s = 0$ and $Y_i^{s'} = 1$. Furthermore, suppose $\hat{Y}_i^s$ and $\hat{Y}_i^{s'}$ are the same, e.g., when considering RETRAIN, then the indicators $\mathbb{1}_{\hat{Y}_i^s = 1}$ and $\mathbb{1}_{\hat{Y}_i^{s'} = 1}$ will be the same and appear in pair (specifically, with opposite sign, one in $\mathrm{Adv}_s$ and another in $\mathrm{Adv}_{s'}$), and hence cancel each other out. This is why the advantage is 0 for RETRAIN. This pairing of indicators for $i$ under splits $s$ and $s'$ happens for imperfect unlearning algorithms, but the indicator might change: $\hat{Y}_i^s$ can swap from 0 to 1 due to imperfect unlearning. If this happens, $i$ contributes $1/3$ to the denominator of the SWAP advantage formula, hence $1/6$ in total (divided by 2 at the end). With this observation, we see that for $\mathrm{UL}_1$, only $B$'s prediction will be flipped from 0 to 1, hence $\mathrm{UL}_1$'s advantage is $1/6$ in the SWAP test. The same argument applies for $\mathrm{UL}_2$ where both $B$ and $C$'s predictions are flipped, hence the advantage is $2/6 = 1/3$. $\qquad\square$

## C  Omitted details from Section 4

### C.1  Computational resource and complexity

We conduct our experiment on `Intel(R) Xeon(R) Gold 6338 CPU @ 2.00GHz` with 4 A40 `NVIDIA` GPUs. It takes approximately 6 days to reproduce the experiment of standard deviation comparison between *SWAP* test and random dataset splitting. It takes approximately 1 day to reproduce the experiment of dataset size and random seeds. Furthermore, it takes approximately 4 days to reproduce the experiment of differential private testing.

### C.2  Details of training

For target model training without differential privacy (DP) guarantees, we consider using the ResNet-20 [He et al., 2016] as our target model and train it with Stochastic Gradient Descent (SGD) [Ruder, 2016] optimizer with a MultiStepLR learning rate scheduler with milestones $[100, 150]$ and an initial learning rate of 0.1, momentum 0.9, weight decay $10^{-5}$. Moreover, we train the model with 200 epochs, and we empirically observe that this guarantees convergence. For a given dataset split, we average 3 models to approximate the randomness induced in training and unlearning procedures.

For training DP models, we use DP-SGD [Abadi et al., 2016] to provide DP guarantees. Specifically, we adopt the OPACUS implementation [Yousefpour et al., 2021] and use ResNet-18 [He et al., 2016] as our target model. The model is trained with the RMSProp optimizer using a learning rate of 0.01 and of 20 epochs. This ensures convergence as we empirically observe that 20 epochs suffice to yield a comparable model accuracy. Considering the dataset size, we use $\delta = 10^{-5}$ and tune the max gradient norm individually.

## C.3 IC score and MIA score

One of the two metrics we choose to compare against is the Interclass Confusion (IC) Test [Goel et al., 2023]. In brief, the IC test "confuses" a selected set of two classes by switching their labels. This is implemented by picking two classes and randomly selecting half of the data from each class for confusion. Then the IC test proceeds to train the corresponding target models on the new datasets and perform unlearning on the selected set using the unlearning algorithm being tested, and finally measures the inter-class error of the unlearned models on the selected set, which we called the *memorization score* $\gamma$. Similar to the advantage, the memorization score is between $[0, 1]$, and the lower, the better since ideally, the unlearned model should have no memorization of the confusion. Given this, to compare the IC test with the Unlearning Quality $\mathcal{Q}$, we consider $1 - \gamma$, and refer to this new score as the *IC score*.

On the other hand, the MIA AUC is a popular MIA-based metric to measure the performance of the unlearning. It measures how MIA performs by calculating the AUC (Area Under Curve) of MIA on the union of the test set and the forget set. We note that AUC is a widely used evaluation metric in terms of classification models since compared to directly measuring the accuracy, AUC tends to measure how well the model can discriminate against each class. Finally, as defined in Section 4, we let the *MIA score* be $1 - $ MIA AUC to have a fair comparison.

**Model Accuracy *versus* DP Budgets.** We also report the classification accuracy of the original model trained with various DP budgets in Table 5. As can be seen, the classification accuracy increases as the $\epsilon$ is relaxed to a larger value, showing the inherent trade-off between DP and utility. For experiments about Unlearning Quality in Table 3 and MIA score in Table 4(B), the original models are shared and thus have the same results (Table 5(B)). We note that measuring the IC scores requires dataset modifications, so the model accuracy in the experiments of IC score (Table 5(A)) differs slightly from that in experiments of Unlearning Quality and MIA score (Table 5(B)).

Table 5: Model accuracy *versus* DP Budgets. See Table 3 for more context.

| (A) Results for experiments in Table 4(A). | | | | | (B) Results for experiments in Tables 3 and 4(B). | | | | |
|---|---|---|---|---|---|---|---|---|---|
| $\epsilon$ | 50 | 150 | 600 | $\infty$ | $\epsilon$ | 50 | 150 | 600 | $\infty$ |
| Accuracy | 0.442* | 0.506* | 0.540* | 0.639* | Accuracy | 0.485* | 0.520* | 0.571* | 0.660* |

**Remark C.1.** *We would like to clarify the performance difference compared to the common literature, which can be attributed to the dataset split. The original dataset is split evenly into the target and shadow datasets for the purpose of implementing MIA. Within the target dataset, further partitioning is performed to create the retain, forget, and test sets. As a result, only about 30% of the full dataset remains available for training the model, significantly reducing the effective training data. Note that the data split is necessary for our experiments, so we cannot get significantly more training data.*

*We experimented with training on the full dataset and applied data augmentation while keeping all other configurations unchanged. With the full dataset, the model achieved an accuracy of 85.34%. After incorporating data augmentation, the accuracy further improved to 91.13%, aligning with the past literature. This simple ablation study validates that the performance difference mainly comes from the difference in data size and the omission of data augmentation.*

*We select a large $\epsilon$ in our differential privacy experiments due to the significant drop in accuracy observed when $\epsilon$ is small, which stems from the same dataset size limitation. We include an additional experiment with a smaller $\epsilon$ in the linear setting, as presented at the end of Appendix C.4.*

## C.4 Additional experiments

In this section, we provide additional ablation experiments on our proposed Unlearning Quality metric by considering varying various parameters and settings.

**Unlearning Quality Versus Model Architecture.** We provide additional experimental results under different model architectures. The experiment is conducted with the CIFAR10 dataset and $\alpha = 0.1$. The results are shown in Table 6. Interestingly, we observe once again that the relative ranking of different unlearning methods stays mostly consistent across different architectures.

Table 6: Unlearning Quality *versus* different model architectures. The relative ranking of different unlearning methods stays mostly consistent under different architectures.

| UL | ResNet44 | ResNet56 | ResNet110 |
|---|---|---|---|
| RETRFINAL | $0.497 \pm 0.040$ | $0.473 \pm 0.010$ | $0.476 \pm 0.036$ |
| FTFINAL | $0.495 \pm 0.041$ | $0.471 \pm 0.011$ | $0.477 \pm 0.039$ |
| FISHER | $0.847 \pm 0.051$ | $0.832 \pm 0.032$ | $0.895 \pm 0.020$ |
| NEGGRAD | $0.562 \pm 0.025$ | $0.537 \pm 0.016$ | $0.520 \pm 0.042$ |
| SALUN | $0.716 \pm 0.008$ | $0.692 \pm 0.013$ | $0.672 \pm 0.033$ |
| SSD | $0.939 \pm 0.053$ | $0.935 \pm 0.056$ | $0.968 \pm 0.017$ |

**Unlearning Quality Versus Dataset.** We provide additional experiments on vision datasets CIFAR100 [Krizhevsky et al., 2009] and MNIST [LeCun, 1998], and natural language dataset SST5 [Socher et al., 2013] . The experiment is conducted with the ResNet20 model architecture and $\alpha = 0.1$ on CIFAR100 and CIFAR10. For SST5 dataset, the experiment is conducted on the BERT model [Devlin et al., 2019] and $\alpha = 0.1$. We note that CIFAR100 has 100 classes and 50000 training images, SST5 has 5 classes and 11855 training sentences while MNIST has 10 classes and 60000 training images. CIFAR100 is considered more challenging than CIFAR10, while MNIST is considered easier than CIFAR10. The results are shown in Table 7.

In this experiment, besides the consistency we have observed throughout this section, we in addition observe that the Unlearning Quality reflects the *level of difficulties* of unlearning on different datasets. Specifically, the Unlearning Quality of most unlearning methods is higher on MNIST while lower on CIFAR100, in comparison to those on CIFAR10 and SST5.

Table 7: Unlearning Quality *versus* different datasets. In addition to the consistency of the Unlearning Quality across unlearning methods, the Unlearning Quality scores are higher on MNIST and SST5 while lower on CIFAR100, in comparison to those on CIFAR10, reflecting the level of difficulties on different datasets.

| UL | CIFAR100 | MNIST | SST5 |
|---|---|---|---|
| RETRFINAL | $0.464 \pm 0.027$ | $0.976 \pm 0.020$ | $0.404 \pm 0.013$ |
| FTFINAL | $0.462 \pm 0.028$ | $0.977 \pm 0.021$ | $0.404 \pm 0.013$ |
| FISHER | $0.606 \pm 0.008$ | $0.990 \pm 0.002$ | $0.552 \pm 0.014$ |
| NEGGRAD | $0.669 \pm 0.016$ | $0.980 \pm 0.017$ | $0.376 \pm 0.017$ |
| SALUN | $0.697 \pm 0.082$ | $0.995 \pm 0.002$ | $0.456 \pm 0.015$ |
| SSD | $0.923 \pm 0.058$ | $0.998 \pm 0.001$ | $0.592 \pm 0.011$ |

**Validation of Theorem 3.5 With Linear Models** We experimented with a method from Guo et al. [2020], which is an unlearning algorithm for linear models with $(\epsilon, \delta)$-certified removal guarantees. We followed the experimental setup in Guo et al. [2020], training a linear model on part of the MNIST dataset for a binary classification task distinguishing class 3 from class 8.

In their algorithm, the parameter $\epsilon$ controls a budget indicating the extent of data that can be unlearned. During the iterative unlearning, when the accumulated gradient residual norm is beyond the unlearning budget, the unlearning guarantee is broken and retraining will kick in. So $\epsilon$ cannot be made arbitrarily small. Below, we report the Advantage metric for their unlearning algorithm with different $\epsilon$ ($\delta$ is fixed as $1e-4$), as well as the Retrain method as reference:

Table 8: Change of advantage in linear models with respect to decreasing $\epsilon$.

| $\epsilon$ | 0.8 | 0.6 | 0.4 | 0.3 | RETRAIN |
|---|---|---|---|---|---|
| Advantage | 0.010 | 0.009 | 0.005 | 0.003 | 0.002 |

We can see that the Advantage monotonically decreases as $\epsilon$ decreases, which aligns with our Theorem 3.5.

