# OpenReview forum: "A Reliable Cryptographic Framework for Empirical Machine Unlearning Evaluation"
_NeurIPS.cc/2025/Conference — NeurIPS 2025 poster_

### Official Review · Reviewer_5sfE · 2025-06-25

**Clarity:** 3
**Significance:** 2
**Originality:** 2
**Rating:** 3
**Confidence:** 3

**Summary:**

This paper introduces a cryptographic game framework based on membership inference attack to evaluate machine unlearning algorithms. It aims to address the unreliability of existing MIA-based evaluation metrics. And the paper proposes a practical tool, the SWAP test, to approximate the proposed evaluation metric. The empirical findings reveal the effectiveness the metric.

**Questions:**

1. Impact of Varying Sensitivity Distribution ($P_D$): The paper defaults to a uniform sensitivity distribution $P_D=U(D)$. What are the theoretical and empirical implications if $P_D$ is non-uniform, reflecting varying data sensitivities or attack focuses? How would this impact the calculated "advantage" and the practical utility of the Unlearning Quality metric?
2. Distinguishing Unlearning Performance under Low Noise/ High Privacy Budgets: As mentioned in weakness 3, in Table 1, for low noise/high privacy budgets, even when the performance differences among the various unlearning algorithms become more pronounced, the Unlearning Quality for various unlearning methods is consistently high, and the differences between different algorithms appear negligible. How then can one compare the advantages and disadvantages of different algorithms?
3. Cross-Dataset/Model Performance Variations and Metric Design: In Appendix C.4, the paper attributes significant performance variations across different datasets and model architectures to models’ learning capabilities. Do other existing MIA-based evaluation methods (e.g., MIA AUC, IC Test) exhibit similar magnitudes of performance differences across these diverse settings, or is this variability particularly pronounced with the proposed metric? If the latter, could this variability be influenced by specific aspects of the evaluation framework itself?
4. Theoretical Bounds for SWAP Test Approximation Error: The paper introduces the SWAP test as a practical approximation method. Are there any theoretical bounds or guarantees on the approximation error of the SWAP test relative to the exact "advantage" metric? Could the authors provide more quantitative analysis regarding the precision of this approximation?
5. Robustness to Advanced MIA Adversaries: If MIA attacks become more powerful, for example, by considering relationships between data and using data covariance for attacks, how could the framework be extended or adapted to incorporate such advanced attack capabilities to ensure its long-term relevance and reliability?

**Ethical Concerns:**

["NO or VERY MINOR ethics concerns only"]

**Limitations:**

1. Assumption of Uniform Data Distribution: The framework relies on a uniform data splitting strategy for the forget and test sets. Deviations from this i.i.d. setup could complicate theoretical guarantees and empirical analysis.
2. Reliance on Weak Adversaries: The current empirical evaluation is predicated on state-of-the-art MIA adversaries being "weak" and making independent decisions per data point, not exploiting data covariance. This might limit the framework's ability to reflect true privacy risks against more sophisticated future attacks.
3. Limited Discernment in High Privacy/Low Noise Regimes: The metric struggles to show significant differentiation between unlearning algorithms when differential privacy is applied, even with larger ϵ values, as all methods achieve very high and similar Unlearning Quality scores
4. Lack of Quantified Approximation Error for SWAP Test: While the SWAP test is proposed as a practical approximation, the paper does not provide theoretical bounds or quantitative analysis of its approximation error for the general "advantage" metric.
5. Variability in Metric Performance Across Diverse Settings: The evaluation metric shows significant variability in performance across different datasets and model architectures. While partly attributed to inherent task difficulty, a deeper discussion on whether this variability is amplified by the evaluation framework itself compared to other metrics is needed.

**Paper Formatting Concerns:**

No issues.

**Quality:**

2

**Strengths And Weaknesses:**

Strengths:
1. Interesting Application of Cryptographic Games: The paper frames the evaluation of unlearning algorithms within the context of cryptographic games, borrowing the concept of "advantage" from cryptography. This approach aims to establish a more reliable and theoretically grounded evaluation metric, which is an interesting knowledge transfer.
2. Practical Feasibility with SWAP Test: Recognizing the computational challenges of exact calculation, the introduction of the SWAP test provides a practical approximation method.
3. Clear Presentation: The paper is logically structured, and the concepts are presented clearly. Weaknesses:
1. Domain Transfer: While the application of cryptographic games to unlearning evaluation is interesting, the core methodological contribution appears to be primarily a transfer and adaptation of existing cryptographic concepts rather than the development of fundamentally new algorithmic techniques for unlearning itself. This could be perceived as more of a modeling contribution than a deep algorithmic innovation for the unlearning process
2. Reliance on "Weak Adversaries": The paper acknowledges that current state-of-the-art MIA adversaries are "weak". If more sophisticated MIAs emerge that can leverage the relationship of data points, the effectiveness of the current evaluation framework in truly reflecting privacy risks might be compromised. The paper states that if weak adversaries improve, their metric would also benefit, but this doesn't alleviate the current limitation.
3. Difficulty in Distinguishing Unlearning Performance under Low Noise/High Privacy Budgets: The paper's experimental results (Table 1) show that when differential privacy is applied (even with a relatively large ϵ, representing a weaker privacy guarantee), the Unlearning Quality for different unlearning methods, including "NONE" (no unlearning) and "REMAIN" (the gold-standard), can be high and quite similar. This implies that the metric might struggle to discern fine-grained differences in unlearning quality with low noise or stringent privacy budgets. However, as the ϵ parameter increases (indicating less stringent privacy), the performance differences among the various unlearning algorithms become more pronounced.

---

> ### Author Rebuttal · Authors · 2025-07-31
>
> We sincerely thank Reviewer 5sfE for taking the time to review our paper and for their constructive feedback and  valuable suggestions of our proposed evaluation framework!:
> > Domain Transfer. This could be perceived as more of a modeling contribution than a deep algorithmic innovation for the unlearning process…
>
> We would like to clarify that our work is intended as an **evaluation framework**, instead of a new unlearning algorithm. The lack of rigorous, reliable metrics for assessing unlearning has been a major obstacle in the field, as also highlighted by the recent NeurIPS unlearning competition. Our cryptographic-game-inspired formulation fills this critical gap by providing a theoretically grounded, practically implementable, and empirically robust evaluation metric for unlearning performance. While we draw upon cryptographic principles, we introduce novel modeling and approximation techniques (e.g., the SWAP test and its instantiation under weak MIAs) tailored for the unlearning setting.
>
> > Reliance on Weak Adversaries. If more sophisticated MIAs emerge that can leverage the relationship of data points, the effectiveness and robustness of the current evaluation framework in truly reflecting privacy risks might be compromised…
>
> We would like to clarify a potential misinterpretation: the reviewer suggested that “If more sophisticated MIAs emerge …, the effectiveness of the current evaluation framework in truly reflecting privacy risks might be compromised”. However, our evaluation framework can in fact be further strengthened when new and stronger MIAs become available.
> As we clarify in Section 3.5 and Appendix B.2, our theoretical framework is fully general—it supports strong adversaries that interact multiple times with the oracle or reason over multiple data points jointly (e.g., with covariance structure). The Unlearning Quality metric is defined as a worst-case guarantee over all efficient adversaries, which naturally includes both weak and strong ones.
> In practice, we adopt weak MIAs in our empirical evaluation because all current state-of-the-art MIAs are weak under our definition—they act independently on each data point. When stronger MIAs become available, our framework can seamlessly accommodate them. In fact, stronger adversaries would only lower the empirical Unlearning Quality, yielding more conservative (and more accurate) estimates of privacy risk. Thus, the flexibility of our metric is a strength, not a limitation.
>
> Nevertheless, we included white-box attacks [2,3] on SST5 dataset with BERT to show that our metric is robust to stronger adversaries. The results are shown below:
> | Method    | 50           | 150          | 600          | ∞             |
> |-----------|--------------|--------------|--------------|---------------|
> | FTFinal   | 0.984 ± 0.001 | 0.960 ± 0.005 | 0.943 ± 0.007 | 0.404 ± 0.013 |
> | RetrFinal | 0.986 ± 0.001 | 0.965 ± 0.006 | 0.944 ± 0.007 | 0.405 ± 0.012 |
> | SalUN     | 0.986 ± 0.002 | 0.980 ± 0.001 | 0.964 ± 0.011 | 0.456 ± 0.015 |
> | Neggrad   | 0.987 ± 0.005 | 0.957 ± 0.007 | 0.932 ± 0.006 | 0.376 ± 0.017 |
> | SSD       | 0.989 ± 0.001 | 0.988 ± 0.006 | 0.965 ± 0.006 | 0.592 ± 0.011 |
> | Fisher    | 0.984 ± 0.002 | 0.982 ± 0.001 | 0.960 ± 0.007 | 0.552 ± 0.014 |
> | None      | 0.983 ± 0.001 | 0.930 ± 0.002 | 0.910 ± 0.004 | 0.368 ± 0.015 |
> | Retrain   | 0.997 ± 0.001 | 0.995 ± 0.003 | 0.991 ± 0.003 | 0.948 ± 0.010 |
>
> We can see that the results remain consistent with those from black-box attacks—the Unlearning Quality is low for the baseline (None), high for exact unlearning (Retrain), and exhibits meaningful variation across approximate methods. This demonstrates that our metric retains its sensitivity and calibration under stronger, white-box adversaries, and further supports the practical robustness and generality of our evaluation framework.
> We will incorporate these results and discussion into the revised version to emphasize that our framework is not limited by the adversary access mode, and can support future advances in attack methodology.
>
> > Difficulty in distinguishing the performance of unlearning algorithms when the model is very private (ε is small) .
>
> We clarify that the lack of significant differentiation between algorithms when ε is small is not a shortcoming, but rather an expected and desirable outcome. When a model is trained with strong differential privacy (small ε), the influence of individual samples is already well-masked, making unlearning inherently easier. In such scenarios, all unlearning methods—whether approximate or exact—perform similarly well, and our metric reflects this reality.
>
> It’s worth noting that this behavior in fact validates and justifies the reliability of our metric: it gives high scores to all methods only when the model is already highly private and thus has little need for aggressive unlearning. As ε increases (weaker DP), we see the expected divergence between algorithms, demonstrating the metric’s sensitivity in regimes where unlearning efficacy truly matters.
>
> > Uniform sensitivity distribution. The framework relies on a uniform data splitting strategy for the forget and test sets. Deviations from this i.i.d. setup could complicate theoretical guarantees and empirical analysis.
>
> We refer to Appendix B.2, where we discuss the role of the sensitivity distribution ​ in shaping oracle responses, as well as the implication of **uniform splitting**. In particular, we highlight that sensitivity distribution models the *privacy risks*, informing the Unlearning Quality to focus/omit particular points with higher probability, while uniform splitting is akin to the standard assumption in the unlearning literature and aligns with the i.i.d. setup of most unlearning setups.
>
> A combination of these two makes our framework flexible, as our framework and theory support arbitrary non-uniform sensitivity distributions to model data sensitivity or adversarial focus. Our experiments consider a uniform sensitivity distribution for simplicity, and exploring this in practice is an exciting direction we plan to pursue in future work, but it is beyond the scope of the current submission.
>
> > Cross-dataset and architecture variability. Do other existing MIA-based evaluation methods (e.g., MIA AUC, IC Test) exhibit similar magnitudes of performance differences across these diverse settings, or is this variability particularly pronounced with the proposed metric?
>
> We would like to point out that the variability observed across datasets and architectures (Appendix C.4) stems primarily from differences in task difficulty and model learning capacity. For example, CIFAR-100 is a harder task than MNIST, and some unlearning methods degrade more noticeably under harder conditions. This variability is expected and also shown in previous literature(e.g., see  Figure 2, Table 5-7 in IC test [1]).
>
> > Theoretical approximation error bounds. Are there any theoretical bounds or guarantees on the approximation error of the SWAP test relative to the exact "advantage" metric?
>
> While we do not currently provide tight theoretical bounds on the approximation error of the SWAP test, we would like to highlight several facts:
>
> 1. The SWAP test is an unbiased estimator of the true advantage, even under arbitrary sensitivity distributions. This is briefly discussed in Appendix B. Hence, under mild assumptions, it can be shown that the expected error between the average of SWAP estimators and the full Unlearning Quality decreases exponentially fast.
> 2. It inherits the zero-grounding property (Proposition 3.6), which is a key theoretical feature of the exact metric.
> Empirically, we observe that the SWAP test preserves the monotonicity with respect to ε and ranking consistency across datasets and methods (Table 1 and C.4).
> Together, these properties suggest that the SWAP test is both principled and reliable, even without explicit error bounds. Formal analysis of its variance and concentration is a promising future direction.
>
> [1] Goel, Shashwat, et al. "Towards adversarial evaluations for inexact machine unlearning." arXiv preprint arXiv:2201.06640(2022).
>
> [2] Nasr, Milad, Reza Shokri, and Amir Houmansadr. "Comprehensive privacy analysis of deep learning: Passive and active white-box inference attacks against centralized and federated learning." 2019 IEEE symposium on security and privacy (SP). IEEE, 2019.
>
> [3] Carlini, Nicholas, et al. "Membership inference attacks from first principles." 2022 IEEE symposium on security and privacy (SP). IEEE, 2022.

---

> ### Author Response · Authors · 2025-08-05
>
> Dear Reviewer 5sfE,
>
> We sincerely thank the reviewer again for the time and efforts in reviewing our paper! As we are coming to the **last three days** of mutual discussion, we kindly hope the reviewer could take a look at our earlier response and let us know any remaining concerns.
>
> Thank you!

---

### Official Review · Reviewer_yVLK · 2025-07-02

**Clarity:** 4
**Significance:** 3
**Originality:** 4
**Rating:** 5
**Confidence:** 4

**Summary:**

This paper proposes a new metric to evaluate the effectiveness of machine unlearning. Machine unlearning is intended to remove the trace of sensitive training data from a model. Existing metrics for unlearning sometimes have counter-intuitive outcome, e.g., assigning low score to models retrained from scratch without the sensitive data. This paper provides a theoretical framework as well as computationally tractable heuristics that instantiate the framework. Empirical evidence shows that the new score is consistent with the expected outcome of DP-guaranteed models across different privacy parameters.

**Questions:**

As the authors mention in the work, one shortcoming of MIA based evaluation is that score may be low for model retrained without the data to be removed. However, is this behavior expected if there are duplicates in the data set? For example, two individuals may have identical data inputs. Will MIA mistake the remained duplicate as the entry to be removed? How does new framework overcome this problem?

**Ethical Concerns:**

["NO or VERY MINOR ethics concerns only"]

**Final Justification:**

The authors have addressed my questions. Therefore, I'm maintaining my positive recommendation for this paper.

**Limitations:**

Yes.

**Quality:**

4

**Strengths And Weaknesses:**

This paper is very well written. The problem setting is clearly motivated. The theoretical framework is concisely and formally defined. The heuristics are sound. The experiment designs are also smart: the authors use different levels of DP to overcome the lack of ground truth effectiveness of unlearning. Overall, I believe this paper is of high quality.

There is no apparent weakness in the work. I do have questions about the expected behavior of unlearning score in some corner cases. I'd appreciate if the authors could clear my doubt.

---

> ### Author Rebuttal · Authors · 2025-07-31
>
> We sincerely thank Reviewer  yVLK for taking the time to review our paper and for their constructive feedback and strong recognition of our proposed evaluation framework!
>
> > Duplicates in datasets.  How does the new framework overcome the case when  two individuals may have identical data inputs. Will MIA mistake the remaining duplicate as the entry to be removed?
>
> This is an intriguing question, but we would argue that, for the purpose of **evaluating** unlearning algorithms, the scenario where the retain set contains duplicate entries of the forget set is a **pathological** evaluation setting. Specifically, the ground truth for whether a model has “forgotten” the sample from the forget set becomes ambiguous.
>
> This pathological setting is analogous to evaluating classification accuracy using mislabeled test samples. When the test samples are mislabeled, it is unclear whether one should expect a better classification algorithm should have higher or lower test accuracy on this corrupted test set. Similarly, it is unclear whether a better unlearning algorithm should have higher or lower Unlearning Quality in the pathological setting where the retain set and forget set share duplicates.

---

> > ### Comment · Reviewer_yVLK · 2025-08-05
> > **Post Rebuttal**
> >
> > I appreciate the clarification and discussion from the authors. My original score was positive and I remain positive after the rebuttal. Thanks!

---

### Official Review · Reviewer_K1Yg · 2025-07-02

**Clarity:** 4
**Significance:** 4
**Originality:** 3
**Rating:** 6
**Confidence:** 4

**Summary:**

This paper proposes a novel, theoretically grounded framework for evaluating machine unlearning algorithms using a game-theoretic approach inspired by cryptographic games. The authors introduce the Unlearning Sample Inference Game, which models the interaction between an unlearning algorithm and a membership inference attack (MIA) adversary. They define a new metric, Unlearning Quality (Q), based on the adversary’s advantage in distinguishing forgotten from unseen data points. The metric satisfies desirable properties such as zero advantage for retraining (i.e., perfect unlearning) and alignment with certified removal guarantees.

To make the framework practical, the authors develop an efficient approximation called the SWAP test, which preserves theoretical guarantees while reducing computational cost. Empirical results on CIFAR-10 demonstrate that the proposed metric is more robust and consistent than existing attack-based evaluation metrics, such as MIA AUC and inter-class confusion, especially under varying differential privacy budgets.

**Questions:**

Have the authors tested the metric’s robustness against white-box MIAs, like gradient-based or loss trajectory attacks? Can the proposed framework incorporate more recent adaptive MIAs (e.g., Carlini’s attacks)?

Do you believe the metric would be stable across different architectures (e.g., transformers, graph models)? Could optimization landscapes affect the adversary’s success, and hence the unlearning score? Another thing is that do you think results would generalize to imbalanced datasets or different modalities? While a more empirical question, would love to hear the authors' thoughts on this.


How sensitive is the unlearning quality metric to non-uniform sensitivity distributions? What happens when the forget and test sets differ in class distribution or sample difficulty?

**Ethical Concerns:**

["NO or VERY MINOR ethics concerns only"]

**Final Justification:**

The authors provide detailed answers  to address my questions as well as provide additional experiments for the adversaries' usecase I highlighted. I am happy with the discussion and high quality attention to detail to address my review.

**Limitations:**

The evaluation framework assumes that the adversary only queries the oracle once, i.e., a “weak adversary,” yet most real-world attackers may use multiple adaptive queries or exploit correlations.

The SWAP test depends on careful dataset splits with equal-sized forget/test sets and uniform sensitivity distribution.

Only black-box or weak MIAs are used, no white-box or shadow-model-free MIAs.

**Quality:**

3

**Strengths And Weaknesses:**

**Strengths:**

1. The paper introduces the Unlearning Sample Inference Game, inspired by cryptographic games, to model the interaction between an unlearning algorithm and an MIA (Membership Inference Attack) adversary. The paper proposes the concept of "advantage" to quantify privacy risk and unlearning efficacy, providing theoretical grounding.

2. The theoretical guarantees provided are strong.
The framework satisfies desirable properties:

- Zero Grounding: The retraining baseline achieves 0 adversarial advantage (i.e., Q = 1).

- Certified Removal Guarantee: Strong connection between the metric and differential privacy–inspired certified removal.

- Handles multiple MIA adversaries, addressing inconsistency in existing MIA-based evaluations.


3. The SWAP test offers an efficient way to approximate the proposed metric while maintaining symmetry and grounding properties. Empirically shown to be more stable than naive split-based MIA evaluations.

4. The paper has comprehensive experiments. The authors evaluate popular unlearning methods (e.g., Fisher Forgetting, SSD, SALUN) on CIFAR-10 with ResNet. They also benchmark performance across varying DP budgets, showing the proposed metric Q aligns well with theoretical expectations.

5. Robustness and consistency are a strength as they compared existing metrics (MIA AUC and Interclass Confusion),
The proposed Unlearning Quality metric maintains monotonic behavior with increasing privacy budget. It also offers a more consistent ranking of unlearning methods across different conditions.

6. The authors also do a thorough review of prior work in attack-based evaluation (e.g., MIA metrics) and theory-based and retraining-based metrics. They effectively position this work as addressing the pitfalls of all three categories using a theoretically justified, attack-based approach.

**Weaknesses / Limitations:**
1. Assumption of Weak Adversaries: Practical evaluation is restricted to weak MIAs (interact only once with the oracle), limiting full assessment of sophisticated attacks.
While justified by current MIA limitations, this could underestimate real-world risks.

2. No Real-World Dataset in that experiments are confined to CIFAR-10, a toy dataset. Lack of tests on large-scale or sensitive datasets (e.g., medical records, text corpora) limits external validity.

3. There seems to be some reliance on certified removal for bounds. Certified removal parameters (ε, δ) are difficult to compute or guarantee in practice, especially for non-convex or large models.

4. Although the SWAP test reduces cost, full computation of the proposed metric can still be computationally expensive in real-world pipelines.

5. While the authors critique other MIA-based methods for giving false confidence, the possibility that their own metric may also be over-optimistic under strong attacks is not discussed in depth.

---

> ### Author Rebuttal · Authors · 2025-07-31
>
> We sincerely thank Reviewer K1Yg for taking the time to review our paper and for their constructive feedback and strong recognition of our proposed evaluation framework!
>
> > Assumption of Weak Adversaries. Practical evaluation is restricted to weak MIAs (interact only once with the oracle), limiting full assessment of sophisticated attacks.
>
> We would like to clarify that while our empirical evaluation leverages weak (single-interaction) MIAs due to the limitations of current state-of-the-art adversaries, our framework is not restricted to weak adversaries in theory. As detailed in Appendix B.2, our cryptographic game formulation accommodates arbitrary adversaries, including those with multi-sample or oracle access (e.g., strong, white-box, or adaptive MIAs).
>
> Indeed, we acknowledge that stronger adversaries may yield lower Unlearning Quality, and thus our reported values may be over-optimistic. However, in practice, users are encouraged to instantiate the strongest MIA feasible in their deployment scenario to obtain a meaningful approximation of worst-case privacy risk.
>
> We will explicitly include this discussion in the revised version to clarify that our metric’s rigor scales with the strength of the employed adversary.
>
> > No Real-World Dataset: Datasets in the experiments are confined to CIFAR-10, a toy dataset.
>
> We want to clarify that in the appendix, we have already included additional datasets (CIFAR-100 and MNIST) to demonstrate robustness across varying dataset complexities.
>
> We agree that incorporating a non-vision modality is important. As such, we are in the process of extending our experiments to include a text corpus (SST5) and pretrained models such as BERT.  The results is shown below:
>
> | Method    | 50           | 150          | 600          | ∞             |
> |-----------|--------------|--------------|--------------|---------------|
> | FTFinal   | 0.986 ± 0.001 | 0.960 ± 0.005 | 0.945 ± 0.008 | 0.425 ± 0.017 |
> | RetrFinal | 0.986 ± 0.001 | 0.967 ± 0.007 | 0.944 ± 0.007 | 0.422 ± 0.020 |
> | SalUN     | 0.991 ± 0.001 | 0.984 ± 0.001 | 0.966 ± 0.008 | 0.525 ± 0.018 |
> | Neggrad   | 0.991 ± 0.004 | 0.961 ± 0.005 | 0.932 ± 0.006 | 0.388 ± 0.013 |
> | SSD       | 0.992 ± 0.004 | 0.991 ± 0.005 | 0.965 ± 0.006 | 0.592 ± 0.011 |
> | Fisher    | 0.990 ± 0.001 | 0.988 ± 0.002 | 0.964 ± 0.006 | 0.564 ± 0.010 |
> | None      | 0.985 ± 0.001 | 0.951 ± 0.004 | 0.929 ± 0.007 | 0.370 ± 0.021 |
> | Retrain   | 0.999 ± 0.001 | 0.995 ± 0.000 | 0.993 ± 0.002 | 0.953 ± 0.008 |
>
>
> We can see that the results also exhibit similar trends as in the vision domain: the proposed metric distinguishes between unlearning methods more clearly as the privacy budget becomes weaker. Moreover, Retrain consistently achieves near-perfect Unlearning Quality, while approximate methods show meaningful variation based on their aggressiveness and design.
>
> > Certified removal parameters are difficult to compute in practice, especially for non-convex or large models.
>
> It is worth noting that we use certified removal only for the purpose of theoretical analysis, connecting our metric to well-established definitions from the literature. This connection provides formal grounding (see Theorem 3.5), but is not required for implementation or evaluation, since the definition of our metric does not rely on certified removal.
>
> > Full computation of the proposed metric can still be computationally expensive
>
> We highlight that our metric has comparable costs to existing MIA evaluation pipelines. Concretely, the cost of the SWAP test is approximately 2× that of a standard MIA pipeline—primarily due to evaluating both swapped splits. Therefore, our methods can be applied to scenarios where existing MIA-based evaluations can be used.
>
> > The possibility that the metric may also be over-optimistic under strong attacks is not discussed.
>
> We agree that our current experimental results may be over-optimistic due to the limitations of available MIAs. We will add a discussion to explicitly acknowledge that Unlearning Quality should be interpreted relative to the strength of the MIA used in practice. In other words, one can always enforce the strongest MIA to date relevant to their application to obtain a practically meaningful measure of Unlearning Quality.
>
> However, we also note that our metric is fundamentally different from existing MIA-performance-based scores: it enjoys zero-grounding under retraining, consistency under differential privacy, and is provably upper-bounded by theoretical guarantees (e.g., Theorem 3.5). In particular, the calibration properties of the proposed metric cannot be achieved by merely strengthening the performance of MIAs.
>
> > White-box MIAs. Will white-box MIAs hurt the robustness of the metric?
>
> While our empirical evaluation used black-box MIAs for efficiency, we stress that our framework accommodates both black-box and white-box adversaries. This is discussed in Appendix B.2. We have added classic white-box attacks as in [1,2] on SST5 dataset with BERT. The results is shown below:
>
> | Method | 50 | 150 | 600 | ∞ |
> |-----------|--------------|--------------|--------------|---------------|
> | FTFinal | 0.984 ± 0.001 | 0.960 ± 0.005 | 0.943 ± 0.007 | 0.404 ± 0.013 |
> | RetrFinal | 0.986 ± 0.001 | 0.965 ± 0.006 | 0.944 ± 0.007 | 0.405 ± 0.012 |
>  | SalUN | 0.986 ± 0.002 | 0.980 ± 0.001 | 0.964 ± 0.011 | 0.456 ± 0.015 |
>  | Neggrad | 0.987 ± 0.005 | 0.957 ± 0.007 | 0.932 ± 0.006 | 0.376 ± 0.017 |
> | SSD | 0.989 ± 0.001 | 0.988 ± 0.006 | 0.965 ± 0.006 | 0.592 ± 0.011 |
> | Fisher | 0.984 ± 0.002 | 0.982 ± 0.001 | 0.960 ± 0.007 | 0.552 ± 0.014 |
>  | None | 0.983 ± 0.001 | 0.930 ± 0.002 | 0.910 ± 0.004 | 0.368 ± 0.015 |
> | Retrain | 0.997 ± 0.001 | 0.995 ± 0.003 | 0.991 ± 0.003 | 0.948 ± 0.010 |
>
> We can see that the results remain consistent with those from black-box attacks—the Unlearning Quality is low for the baseline (None), high for exact unlearning (Retrain), and exhibits meaningful variation across approximate methods. This demonstrates that our metric retains its sensitivity and calibration under stronger, white-box adversaries, and further supports the practical robustness and generality of our evaluation framework.
>
> We will incorporate these results and discussion into the revised version to emphasize that our framework is not limited by the adversary access mode, and can support future advances in attack methodology.
>
> > Generalization across architectures, modalities, or imbalanced datasets.
>
> It’s worth noting that our framework is model-agnostic and modality-agnostic by design. The cryptographic game only assumes an unlearned model and a forget/test split—it is the MIA’s task to adapt its attack strategy to the data distribution and architecture. Nevertheless, we have added the results of BERT finetuned on SST5. Please refer to the “No real-world dataset” section for experimental results.
>
> > Non-uniform sensitivity distributions
>
> We have provided a high-level discussion in Appendix B.2, where we define the sensitivity distribution \$\mathbb{P}_{\mathcal{D}}\$ and allow arbitrary biases toward specific samples. This enables the metric to reflect heightened privacy risk from more sensitive data points, addressing more difficult samples, or accommodating class imbalances.
>
> Our design ensures that such distributions are respected during evaluation via the oracle sampling. As a result, the metric captures the influence of both distributional skew and sample difficulty, offering a more realistic view of privacy risk than uniform sampling alone.
>
> [1] Nasr, Milad, Reza Shokri, and Amir Houmansadr. "Comprehensive privacy analysis of deep learning: Passive and active white-box inference attacks against centralized and federated learning." 2019 IEEE symposium on security and privacy (SP). IEEE, 2019.
>
> [2] Carlini, Nicholas, et al. "Membership inference attacks from first principles." 2022 IEEE symposium on security and privacy (SP). IEEE, 2022.

---

> > ### Comment · Reviewer_K1Yg · 2025-08-04
> >
> > Thanks for your detailed answers as well as the additional experiments. I am happy with your response and high quality attention to detail to address my review. I have increased my score.

---

> > > ### Author Response · Authors · 2025-08-04
> > >
> > > Thank you so much for acknowledging our response and updating the rating. We really appreciate your constructive feedback and recognition of our work!

---

### Official Review · Reviewer_FFru · 2025-07-04

**Clarity:** 3
**Significance:** 2
**Originality:** 2
**Rating:** 4
**Confidence:** 3

**Summary:**

This work introduces a notion of unlearning evaluation defined via a distinguishing game dubbed the “unlearning sample inference game.” Informally, this game asks an adversary to distinguish unlearned data from data never used in training at all, given access to the model. The distinguishing game is presented in the style of cryptographic security games, where the adversary is assumed to be computationally bounded and the quality of unlearning is described as a function of the adversary’s distinguishing advantage. Notably, the “unlearning” method of retraining a model on a dataset from which all unlearned data has been removed guarantees that any adversary will have no advantage at all in the distinguishing game. A test for evaluating the advantage of the adversary without computing it precisely is proposed and empirically evaluated on the CIFAR10 dataset and ResNet model output by LR. Retraining, Fisher forgetting, final layer fine-tuning, final layer retraining, negative gradient descent, saliency unlearning, selective synaptic dampening, and no unlearning are evaluated.

**Questions:**

See comments on strength and weaknesses

**Ethical Concerns:**

["NO or VERY MINOR ethics concerns only"]

**Final Justification:**

My appraisal of the paper is positive and I believe the comparison to recent related work will improve the paper. I am maintaining my score.

**Limitations:**

Yes

**Quality:**

3

**Strengths And Weaknesses:**

The distinguishing framework for unlearning evaluation is very natural and I believe captures a very reasonable notion of unlearning. I have some questions about the usefulness of this definition compared to other potential definitions, however. It would be nice to better understand how this definition compares to one where the adversary is computationally unbounded. There have been challenges in separating computational differential privacy from standard (statistical) differential privacy, for instance, and so I’m not sure I appreciate why restricting to computationally bounded adversaries is the right thing to do here, and how this changes the threat model. Some discussion of this restriction would help with interpreting the definition.

There is significant overlap between this paper and another recent paper [1] up on arxiv, so I’d recommend citing this work as well. I believe this went up after the NeurIPS abstract deadline, however, so I don’t think there’s an obligation to have compared to the work already.

[1] Mirror Mirror on the Wall, Have I Forgotten it All? A New Framework for Evaluating Machine Unlearning. Brennon Brimhall, Philip Mathew, Neil Fendley, Yinzhi Cao, Matthew Green

---

> ### Author Rebuttal · Authors · 2025-07-31
>
> We sincerely thank Reviewer FFru for taking the time to review our paper and for their constructive feedback and recognition of our proposed evaluation framework!
> > On the use of computationally bounded adversaries
>
> We would like to clarify that our theoretical framework **does not require** the adversary to be computationally bounded. In fact, **Theorem 3.5 holds even for computationally unbounded adversaries**, meaning our Unlearning Quality metric can naturally generalize to a “statistical” notion without needing to assume computational limitations (see main paper, Theorem 3.5 and Appendix B.3). Thus, the core theoretical foundation of our framework remains valid in both settings.
> That said, we explicitly focus on **computationally bounded adversaries for practical purposes**. As with differential privacy, empirical evaluation and implementation inherently require working with efficient algorithms. Moreover, existing MIA techniques are themselves limited in computational power, and our approximation methods (e.g., the SWAP test) are grounded in this practical setting. In this way, **the use of weak or computationally bounded adversaries serves as a realistic approximation for empirical evaluation**—analogous to the use of efficient distinguishers in applied cryptography.
> We have updated the discussion in the main text to better articulate the above tradeoff between theoretical generality and practical feasibility. We thank the reviewer again for prompting us to clarify this important point.
>
> > "There is significant overlap between this paper and another recent paper [1] up on arXiv..."
>
> We will cite it in the updated version of our paper. As noted, the timing suggests the work appeared after the NeurIPS abstract deadline, but we agree it is important to acknowledge concurrent developments and position our contributions accordingly.

---

> > ### Comment · Reviewer_FFru · 2025-08-06
> >
> > Thank you to the authors for their clarifications, I will maintain my score.

---

### Note · Authors · 2025-08-13

Dear Area Chair and Reviewers,

We sincerely thank the reviewers for their time and efforts in reviewing our paper! Here, we wish to highlight our key contributions, as recognized across the reviews:
- **Novel Theoretical Framework (all reviewers):** We formalize a **novel** *Unlearning Sample Inference Game*, introducing a theoretically grounded evaluation metric for data removal efficacy.
- **Strong Theoretical Guarantees (all reviewers):** We establish several provable properties of the metric, addressing pitfalls of existing MIA-based metrics; Reviewer FFru further noted that our formulation is *“very natural and reasonable.”*
- **Practical and Efficient Approximation (K1Yg, 5sfE)**: We propose the SWAP test as a simple yet effective method for efficient empirical evaluation without loss of rigor.
- **Convincing Empirical Validation (K1Yg, yVLK)**: Extensive experiments confirm that the SWAP test is robust and reliable, preserving the metric’s key theoretical advantages.

During rebuttal, we addressed each reviewer’s concerns as follows:
- **Reviewer FFru’s concerns on computationally bounded adversaries:** We clarified our *Unlearning Quality* metric is fully general, and that we focus on bounded ones only for practical evaluation.
- **Reviewers K1Yg and 5sfE’s concerns on adversary strength:** While SOTA MIAs are all weak under our definition, our framework supports strong ones and remains robust with white-box MIAs, shown empirically.
- **Reviewer K1Yg’s concerns on certified removal, computational cost, and dataset diversity & Reviewer 5sfE’s concerns on high-privacy regimes, sensitivity distributions, and SWAP test reliability:** We provided clarifications, theoretical justifications, confirming our metric remains sound in theory, relevant in practice, and validated across diverse settings. Empirically, we **extended experiments to BERT & SST5, demonstrating consistent robustness across modalities**.

After the rebuttal, we are glad to see that **Reviewers FFru, K1Yg, and yVLK positively acknowledged our response**, with **Reviewer K1Yg explicitly increasing their score**. While we unfortunately didn't hear back from Reviewer 5sfE, we believe our clarifications, additional experiments, and extended discussions have comprehensively addressed the reviewer’s concerns. We hope the AC and the reviewers could leverage our response and this final remark in making the final decision.

Thank you for your time and constructive feedback!

Sincerely,

Authors

---

### Decision · Program_Chairs · 2025-09-17

**Decision:**

Accept (poster)

**Comment:**

This paper introduces a theoretically grounded framework for evaluating machine unlearning through a cryptography-inspired game-theoretic formulation called the Unlearning Sample Inference Game. In this game, an adversary attempts to distinguish between data that has been unlearned and data never used in training, with the quality of unlearning defined by the adversary’s distinguishing advantage. The authors propose a new metric, Unlearning Quality (Q), which satisfies desirable properties such as assigning zero advantage to retraining (i.e., perfect unlearning) and aligning with certified removal guarantees.

To make evaluation practical, they developed the SWAP test, an efficient approximation that retains theoretical guarantees while reducing computational overhead. Empirical results on CIFAR-10 with ResNet show that this metric is robust and consistent across a range of unlearning methods—including retraining, Fisher forgetting, fine-tuning, and gradient-based approaches—and avoids counterintuitive outcomes observed with existing metrics such as MIA AUC or inter-class confusion, particularly under different differential privacy budgets.

The reviewers agreed on the paper’s technical contributions, noting the theoretically sound results, practical and efficient approximations, and extensive empirical validations.